# SNIP: Bridging Mathematical Symbolic and Numeric Realms with Unified Pre-training

**Kazem Meidani**[*1] , **Parshin Shojaee**[*2] , **Chandan K. Reddy** [2] , **Amir Barati Farimani** [1,3]
[1] Department of Mechanical Engineering, Carnegie Mellon University
[2] Department of Computer Science, Virginia Tech
[3] Machine Learning Department, Carnegie Mellon University

## Abstract

In an era where symbolic mathematical equations are indispensable for modeling complex natural phenomena, scientific inquiry often involves collecting observations and translating them into mathematical expressions. Recently, deep learning has emerged as a powerful tool for extracting insights from data. However, existing models typically specialize in either numeric or symbolic domains, and are usually trained in a supervised manner tailored to specific tasks. This approach neglects the substantial benefits that could arise from a task-agnostic multi-modal understanding between symbolic equations and their numeric counterparts. To bridge the gap, we introduce SNIP, a Symbolic-Numeric Integrated Pre-training model, which employs contrastive learning between symbolic and numeric domains, enhancing their mutual similarities in the embeddings. By performing latent space analysis, we observe that SNIP provides cross-domain insights into the representations, revealing that symbolic supervision enhances the embeddings of numeric data and vice versa. We evaluate SNIP across diverse tasks, including symbolic-to-numeric mathematical property prediction and numeric-to-symbolic equation discovery, commonly known as symbolic regression. Results show that SNIP effectively transfers to various tasks, consistently outperforming fully supervised baselines and competing strongly with established task-specific methods, especially in the low data regime scenarios where available data is limited [1].

## 1 Introduction

Throughout the history of science, symbolic mathematics has been unreasonably effective in representing natural phenomena (Wigner, 1960). Complex patterns of natural systems, represented as numeric data observations, can be elegantly abstracted using mathematical formulas. Mathematical symbolism has given us the language to describe, understand, and predict the natural world. The challenge of bridging the gap between the numeric observations and their mathematical symbolic representations has been a consistent focus in many scientific and engineering domains. Recognizing and exploring this connection is crucial, as it promises to drive advancements in various fields.

In recent years, deep learning has demonstrated promising capabilities in learning from symbolic mathematics language as well as extracting insights from numeric data observations. Transformer models (Vaswani et al., 2017), in particular, have emerged as frontrunners in this field, effectively capturing patterns within mathematical expressions and solving complex tasks such as differential equations and function integration (Lample & Charton, 2020; Welleck et al., 2022). However, these models, while powerful, are not inherently designed to handle numeric data inputs. While some pretrained symbolic regression models have been introduced to map numeric datasets to their governing mathematical expressions in a supervised manner (Biggio et al., 2021; Kamienny et al., 2022), a gap still remains in developing a task-agnostic pre-training model capable of mutual understanding between the modalities of symbolic mathematical equations and their corresponding numeric data.

Multi-modal pre-training models, exemplified by groundbreaking models like Contrastive Language-Image Pre-training (CLIP) (Radford et al., 2021), have found a significant place in the

---

*Equal Contribution. Contact email: mmeidani@andrew.cmu.edu

[1]Code and model are available at: `https://github.com/deep-symbolic-mathematics/Multimodal-Math-Pretraining`

deep learning landscape. CLIP has particularly set new standards in vision-language tasks, bridging the understanding between visual content and natural language descriptions. Expanding beyond traditional vision-language domains, recent studies have broadened multi-modal pre-training to include other modalities, such as audio and tabular data (Liu et al., 2021; Zhang et al., 2023; Hager et al., 2023). Additionally, previously untouched scientific domains, like molecular representation, are also benefiting from these advancements in multi-modal representations (Su et al., 2022; Cao et al., 2023). Nevertheless, the symbolic-numeric domain remains relatively unexplored. Considering the foundational role of symbolic mathematics in science and the ubiquity of numeric data, an in-depth exploration of their mutual learning is not only timely but essential.

In this work, we present **S**ymbolic-**N**umeric **I**ntegrated **P**re-training (**SNIP**) to connect the two often distinct worlds of symbolic mathematical expressions and their corresponding numeric manifestations. The architecture of SNIP, depicted in Fig. 1, incorporates dual Transformer encoders, with each encoder dedicated to learning the symbolic or numeric representations of mathematical functions. Subsequently, a task-agnostic contrastive objective is employed to enhance the similarity between (symbolic, numeric) pairs of data. The multi-modal pre-training of SNIP provides capabilities to understand and generate cross-modal content. Our experiments show that SNIP achieves remarkable performance in cross-modal mathematical understanding and prediction tasks. Additionally, by combining SNIP with an equation generation decoder and exploiting its interpolatable latent space, we can effectively harness SNIP's mutual knowledge for the task of numeric-to-symbolic equation discovery (known as symbolic regression), achieving competitive results with state-of-the-art baselines. The major contributions of this work can be summarized as follows:

- Proposing SNIP, a pioneering pre-training method that integrates mathematical symbolic and numeric domains through joint representation learning. This approach captures mutual relationships, delivering embeddings that are informed and enhanced by both domains.

- Evaluating SNIP in cross-modal comprehension across different mathematical property prediction tasks. Our results indicate that SNIP outperforms the fully supervised baselines, particularly in low data regime scenarios. Visualizing the latent embeddings also confirms that SNIP's pre-trained representations reveal patterns linked to these cross-modal mathematical properties.

- Leveraging SNIP for numeric-to-symbolic equation generation task, commonly known as symbolic regression. In this task, after training an expression generation decoder on top of SNIP's numeric encoder, we exploit the high-quality semantic within SNIP's continuous and low-dimensional latent representations to perform latent space optimization with the objective of finding equations with balanced accuracy-complexity. Results show that SNIP achieves state-of-the-art performance on the well-known SRBench (La Cava et al., 2021) benchmark.

## 2 RELATED WORK

**Large-scale Pre-training.** Our work is built upon an extensive body of research advocating the advantages of pre-training large models on large datasets (Zhou et al., 2023; Zong et al., 2023). Initially, pre-training was single-modal, with self-supervised learning (SSL) as a key paradigm that used data as its own supervision, especially useful where labeled data was limited (Balestriero et al., 2023). This paved the way for the emergence of multi-modal pre-training, where models are trained to understand relationships across different modalities (Wang et al., 2023). Vision and language have traditionally played the two main characters of pre-training models. For instance, CLIP (Radford et al., 2021), ALIGN (Jia et al., 2021), and FLAVA (Singh et al., 2022) utilize image-caption pairs to construct jointly learned embedding spaces. These models are trained to align the embeddings of corresponding image-caption pairs while distancing unrelated pairs. The success of multi-modal pre-training in vision and language spurred its adoption in other domains. For example, recent works have extended this approach to videos, audio, and even tabular data (Liu et al., 2021; Dong et al., 2022; Hager et al., 2023). Specialized scientific domains have also embraced this paradigm. For instance, different models have emerged to learn joint representations of molecules (Su et al., 2022; Cao et al., 2023). Our work introduces a fresh perspective, intertwining symbolic mathematics with numeric observations. To this end, we use multi-modal pre-training's potential to deepen the symbolic-numeric mutual understanding.

**Deep Symbolic Mathematics.** Recently, deep learning models have made significant performance in the field of mathematical reasoning (Saxton et al., 2019; Lu et al., 2023). The Transformer models, originally designed for NLP tasks (Vaswani et al., 2017), have been repurposed with remarkable success in the domain of symbolic mathematics. It has powered models that can integrate functions (Lample & Charton, 2020; Welleck et al., 2022), prove mathematical theorems (Lample et al.,

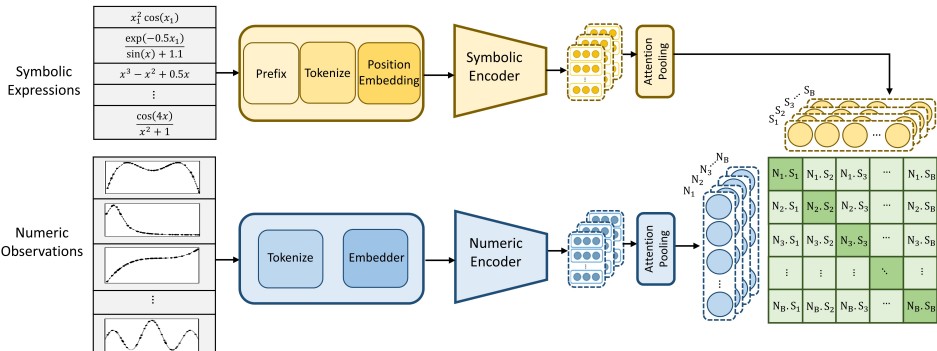

Figure 1: The SNIP Framework: A schematic representation of the dual-encoder pre-training scheme for mutual learning between symbolic equations and their numerical observations. Both symbolic and numeric encoders work in tandem, capturing the paired similarities and essence of their respective modalities.

2022), and perform numerical calculations, such as arithmetic operations (Charton, 2022; Jelassi et al., 2023). These achievements underscore the flexibility and potential of deep learning models in abstract reasoning. Beyond pure symbolic reasoning, there is also a growing interest in supplementing these models with numerical knowledge for improved mathematical understanding. For example, recent works have studied to enhance language models with numeric representations, aiming to improve their skills in mathematical word problem-solving (Peng et al., 2021; Liang et al., 2022; Thawani et al., 2021; Geva et al., 2020). Some recent studies have also explored different strategies for tokenizing and encoding numeric data, such as using multi-hot or continuous representation of numbers (Biggio et al., 2021; Becker et al., 2023; Golkar et al., 2023). Our work contributes a new angle to this growing field by integrating symbolic and numeric understanding in a unified multi-moal pre-training framework. By doing so, we not only capture the abstract representations of mathematical symbolic concepts but also their tangible numeric behaviors.

**Symbolic Regression.** Symbolic regression (SR) concentrates on discovering mathematical expressions for complex systems and representing data patterns in interpretable symbolic form. It has broad implications in both science and engineering, facilitating the modeling of diverse physical phenomena (Cranmer et al., 2020; Rudy et al., 2017; Meidani & Barati Farimani, 2023). Genetic Programming (GP) algorithms laid the foundation for SR, offering methods to search the vast space of mathematical expressions (Schmidt & Lipson, 2009; Cranmer, 2023). The ascent of deep learning subsequently gave rise to neural network-centric methods to reinforce SR's representational capabilities (Petersen et al., 2021). Some pioneering works also combined the evolutionary strengths of GP with the adaptability of neural networks, aiming for a better SR search (Udrescu & Tegmark, 2020; Mundhenk et al., 2021). However, these methods often struggle with challenges such as computational intensity, limited semantic depth, and the necessity to reinitiate search for different datasets. Inspired by the success of pre-trained Transformers in NLP, recent works introduced pre-trained models for SR (Biggio et al., 2021; Kamienny et al., 2022; Shojaee et al., 2023), using synthetic data and pre-trained priors for equation generation. Our multi-modal pre-trained model, SNIP, advances this research towards a more insightful SR direction, leveraging rich encodings that harmoniously bridge symbolic equations with their numeric counterparts.

## 3 PRE-TRAINING

As depicted in Fig. 1, the SNIP architecture comprises two Transformer encoders, each tailored for learning the symbolic or numeric representations of mathematical functions. These symbolic and numeric encoders are jointly trained with a task-agnostic contrastive learning objective to predict correct pairings within a batch of (symbolic, numeric) examples. During pre-training, SNIP receives synthetically created symbolic equations and their associated numeric data as inputs to the symbolic and numeric heads, respectively. In total, SNIP is pre-trained on approximately 60 million synthetic paired examples.

### 3.1 NUMERIC ENCODER

The numeric encoder's foundation is rooted in the recent advancements of Transformer models for encoding numeric observations into latent spaces (Kamienny et al., 2022; D'Ascoli et al., 2022; Biggio et al., 2021). In this framework, the numeric encoder—represented as $\mathcal{E}_\theta^V$—integrates an

embedder, a multi-layer Transformer, and an attention pooling approach, to map numeric observations $(\boldsymbol{x}, \boldsymbol{y})$ into a condensed latent vector $\boldsymbol{Z}_V$.

**Tokenization.** Following (Charton, 2022; Kamienny et al., 2022), numeric inputs are tokenized using base-10 floating-point notation. They are rounded to four significant digits and subsequently represented as sequences of three tokens: sign, mantissa (0-9999 range), and exponent ($E$-100 to $E$100). For instance, the number $5.432$ is tokenized as $[+, 5432, E\text{-}3]$.

**Encoding.** Given a batch of $N$ numeric input points $(\boldsymbol{x}, \boldsymbol{y}) \in \mathbb{R}^{D+1}$, each is represented by $3(D + 1)$ tokens. With increasing $D$ and $N$, the input sequence length grows, challenging the quadratic complexity of Transformers. To address this, we employ an embedder, as suggested by (Kamienny et al., 2022), before the Transformer encoder. This embedder maps each input point to a unique embedding space. The resulting embeddings, with dimension $d_{\text{emb}}$, are then fed into the encoder. For the numeric encoder, we utilize a multi-layer Transformer architecture (Vaswani et al., 2017). Notably, due to the permutation invariance of the $N$ input points for each batch sample, we exclude positional embeddings, aligning with the approach in (Biggio et al., 2021). This encoder variant is denoted as $Enc^V$. The representation at its $l$-th layer is given by $\boldsymbol{V}_l = Enc_l^V(\boldsymbol{V}_{l-1})$, where $l$ ranges from 1 to $L_V$, and $L_V$ signifies number of layers within the numeric encoder.

**Attention-based Distillation.** To distill the information from the Transformer's output into a compact representation for the whole sequence of observations, we employ an attention-based pooling mechanism, following (Santos et al., 2016). Let $\boldsymbol{\mathcal{A}}_V$ denote the attention weights, which are computed as: $\boldsymbol{\mathcal{A}}_V = \text{softmax}\left(\boldsymbol{W}_a \cdot \boldsymbol{V}_{L_V}^T\right)$, where $\boldsymbol{W}_a \in \mathbb{R}^{d_{\text{emb}}}$ is a learnable weight matrix, and we take the transpose of $\boldsymbol{V}_{L_V} \in \mathbb{R}^{N \times d_{\text{emb}}}$ to apply softmax along the sequence dimension $N$. The compact sequence-level representation, $\boldsymbol{Z}_V$, is then obtained by: $\boldsymbol{Z}_V = \boldsymbol{\mathcal{A}}_V \cdot \boldsymbol{V}_{L_V}$. This attention mechanism allows the model to focus on the most informative parts of the data points, effectively compressing the information into a fixed-size embedding.

## 3.2 SYMBOLIC ENCODER

The symbolic encoder in our framework also draws inspiration from recent advancements in Transformer models for encoding symbolic mathematical functions, as demonstrated in works such as (Welleck et al., 2022; Lample & Charton, 2020). Here, the symbolic encoder—denoted as $\mathcal{E}_\psi^S$—is a composite entity parameterized by $\psi$, encapsulating the embedder, a multi-layer Transformer, and attention-based pooling mechanisms. Given an input symbolic expression $f$, this encoder outputs a condensed representation $\boldsymbol{Z}_S$.

**Tokenization.** Mathematical expressions are tokenized by prefix order of their trees, following the principles outlined in (Lample & Charton, 2020). This process employs self-contained tokens to represent operators, variables, and integers, while constants are encoded using the same methodology as discussed in Sec. 3.1, representing each with three tokens. In alignment with (Lample & Charton, 2020), we use special tokens $[\langle BOS \rangle]$ and $[\langle EOS \rangle]$ to mark sequence start and end.

**Encoding.** Given a batch of symbolic expressions with $M$ tokens, each symbolic input is represented as $\boldsymbol{S}_0 = \left[\boldsymbol{E}_{[\langle BOS \rangle]}; \boldsymbol{E}_{t_1}; \ldots; \boldsymbol{E}_{t_M}; \boldsymbol{E}_{[\langle EOS \rangle]}\right] + \boldsymbol{S}^{pos}$, where $\boldsymbol{S}_0 \in \mathbb{R}^{(M+2) \times d_{\text{emb}}}$. Here, $\boldsymbol{E}$ refers to the embedding matrix, $t_i$ denotes the $i$-th token, $M$ signifies the number of tokens in the symbolic expression, $d_{\text{emb}}$ is the embedding dimension, and $\boldsymbol{S}^{pos}$ represents the positional embedding matrix. In the symbolic encoder, we use a Transformers model with the same architecture as in Sec. 3.1. This variant of the encoder, denoted as $Enc^S$, processes the symbolic inputs. The $l$-th layer representation is described as $\boldsymbol{S}_l = Enc_l^S(\boldsymbol{S}_{l-1})$, where $l$ varies from 1 to $L_S$, and $L_S$ indicates number of layers within the symbolic encoder.

**Attention-based Distillation.** The symbolic encoder also employs attention-based pooling, as in Sec. 3.1. This mechanism computes weighted sums to distill information from the symbolic expression into a compact representation $\boldsymbol{Z}_S = \boldsymbol{\mathcal{A}}_S \cdot \boldsymbol{S}_{L_S}$, using attention weights $\boldsymbol{\mathcal{A}}_S$ through softmax along the symbolic sequence.

## 3.3 UNIFIED PRE-TRAINING OBJECTIVE

Our work introduces a multi-modal symbolic-numeric pre-training approach, SNIP, which aims to facilitate a mutual understanding of both domains, enabling advanced cross-modal reasoning.

**Training Objective.** SNIP's pre-training objective is inspired by the joint training used in CLIP (Radford et al., 2021). Incorporating both a numeric and symbolic encoder, the model optimizes a symmetric cross-entropy loss over similarity scores. It employs a contrastive loss (InfoNCE (Oord et al., 2018) objective) to learn the correspondence between numeric and symbolic data pairs. Specif-

ically, this approach learns to align embeddings of corresponding symbolic-numeric pairs while distancing unrelated pairs. The objective function can be defined as:

$$\mathcal{L} = - \sum_{(v,s) \in B} \big( \log \text{NCE}(\boldsymbol{Z}_S, \boldsymbol{Z}_V) + \log \text{NCE}(\boldsymbol{Z}_V, \boldsymbol{Z}_S) \big), \tag{1}$$

where $B$ represents the batch of (symbolic, numeric) data pairs, $\text{NCE}(\boldsymbol{Z}_S, \boldsymbol{Z}_V)$ and $\text{NCE}(\boldsymbol{Z}_V, \boldsymbol{Z}_S)$ denote the contrastive losses on symbolic-to-numeric and numeric-to-symbolic similarities, respectively. The symbolic-to-numeric contrastive loss, $\text{NCE}(\boldsymbol{Z}_S, \boldsymbol{Z}_V)$, is calculated as:

$$\text{NCE}(\boldsymbol{Z}_S, \boldsymbol{Z}_V) = \frac{\exp\left(\boldsymbol{Z}_S \cdot \boldsymbol{Z}_V^+\right)}{\sum_{\boldsymbol{Z} \in \{\boldsymbol{Z}_V^+, \boldsymbol{Z}_V^-\}} \exp\left(\frac{\boldsymbol{Z}_S \cdot \boldsymbol{Z}}{\tau}\right)}. \tag{2}$$

where $\tau$ is temperature, $\boldsymbol{Z}_V^+$ represents positive SNIP numeric embeddings that overlap with SNIP symbolic embedding $\boldsymbol{Z}_S$, and $\boldsymbol{Z}_V^-$ are negative numeric embeddings implicitly formed by other numeric embeddings in the batch. A symmetric equivalent, $\text{NCE}(\boldsymbol{Z}_V, \boldsymbol{Z}_S)$, also defines the numeric-to-symbolic contrastive loss. More implementation details are provided in App. B.

### 3.4 PRE-TRAINING DATA

In our SNIP approach, pre-training relies on a vast synthetic dataset comprising paired numeric and symbolic data. We follow the data generation mechanism in (Kamienny et al., 2022), where each example consists of $N$ data points $(x, y) \in \mathbb{R}^{D+1}$ and a corresponding mathematical function $f$, where $y = f(x)$. Data generation proceeds in several steps, ensuring diverse and informative training examples. More details about each of the following steps are provided in App. A.

**Sampling of functions.** We create random mathematical expressions using a process detailed in (Kamienny et al., 2022; Lample & Charton, 2020). This process involves selecting an input dimension $D$, determining the number of binary operators, constructing binary trees, assigning variables to leaf nodes, inserting unary operators, and applying random affine transformations. This method ensures a diverse set of functions for training.

**Sampling of datapoints.** After generating a function, we sample $N$ input points and find their corresponding target values. To maintain data quality, we follow guidelines from (Kamienny et al., 2022), discarding samples with inputs outside the function's domain or exceptionally large output values. Our approach includes drawing inputs for each expression from various distributions, enhancing training diversity. The generation process of datapoints also involves selecting cluster weights and parameters, sampling input points for each cluster, and normalization along each dimension. To emphasize on the function's numeric behavior rather than the range of values, we also normalize the target values $\boldsymbol{y}$ between $(0, 1)$.

## 4 USING SNIP FOR CROSS-MODAL PROPERTY PREDICTION

To evaluate SNIP's capability for cross-modal comprehension between symbolic and numeric domains, we conducted targeted experiments. These tests aimed to assess the model's aptitude for predicting specific numeric mathematical properties based on the symbolic inputs—a non-trivial task requiring mutual understanding of both domains. For this purpose, we identified a set of mathematical properties; details can be found in App. C. In this section, we focus on two numeric properties for one-dimensional datasets: *Non-Convexity Ratio (NCR)*, and Function *Upwardness*. The *NCR* approximates function convexity with values between `NCR=0` (fully convex) and `NCR=1` (fully concave). *Upwardness* quantifies the function's directionality by assessing the segments where data increases within the training domain, ranging from `UP=-1` for strictly decreasing functions to `UP=1` for increasing ones. Due to space limitations, only results for *NCR* and *Upwardness* are discussed here. A complete list of properties with their detailed prediction results and corresponding chance levels, as well as their SNIP's pre-trained representations, are provided in App. C.

### 4.1 MODELS AND TRAINING

To assess property prediction on top of SNIP's embeddings, we employ a predictor head that passes these embeddings through a single-hidden-layer MLP to yield the predicted values. We adopt a Mean Squared Error (MSE) loss function for training on continuous properties. We consider three key training configurations to probe the efficacy of SNIP's learned representations:

• **Supervised Model**: Utilizes the same encoder architecture as SNIP but initializes randomly.
• **SNIP (frozen)**: Keeps the encoder weights fixed, training only the predictor head.
• **SNIP (finetuned)**: Initializes encoder from pretrained SNIP, allowing full updates during training.

For a fair comparison, all model variants are trained on identical datasets comprising 10K equations and subsequently tested on a distinct 1K-equation evaluation dataset. These datasets are generated using the technique described in Sec. 3.4.

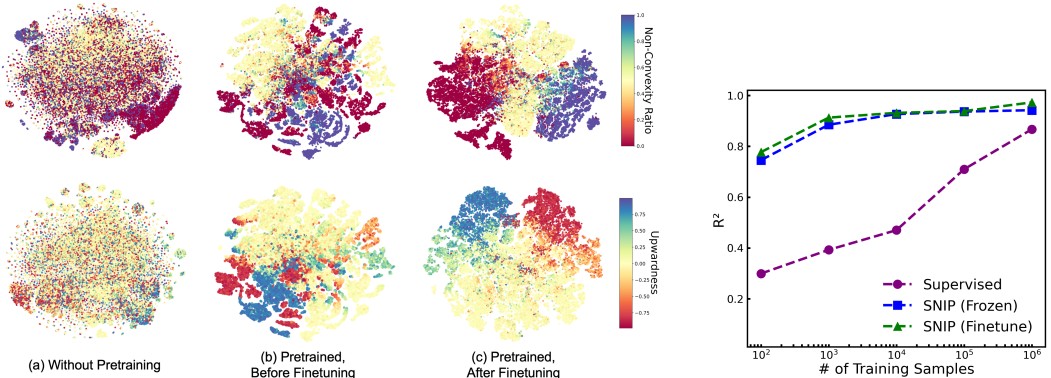

Figure 2: 2D t-SNE representations of the encoded vectors across three model variants, colored for **(top)** Non-Convexity Ratio and **(bottom)** Function Upwardness prediction tasks.

Figure 3: $R^2$ scores for *NCR* property prediction task vs. the number of training samples.

## 4.2 RESULTS

**Quantitative Results.** Table 1 presents the Normalized Mean Squared Error (NMSE) and accuracy metric $Acc_{0.1}$ for all three models across the tasks of predicting *NCR* and *Upwardness*. Here, $Acc_{0.1}$ reflects the percentage of predictions within absolute tolerance $\tau = 0.1$ of

Table 1: Results of using SNIP for property prediction.

| Model | Non-Convexity Ratio | | Upwardness | |
|---|---|---|---|---|
| | $\downarrow$ NMSE | $\uparrow Acc_{0.1}$ | $\downarrow$ NMSE | $\uparrow Acc_{0.1}$ |
| Supervised | 0.5299 | 0.565 | 0.5356 | 0.563 |
| SNIP (frozen) | 0.0731 | 0.861 | 0.0540 | 0.847 |
| SNIP (finetuned) | **0.0683** | **0.921** | **0.0400** | **0.901** |

the true normalized values: $Acc_{\tau} = \frac{1}{N_{test}} \sum_i \mathbb{1} \{|\hat{p}_i - p_i| \leq \tau\}$ where $p_i$ and $\hat{p}_i$ are the true and predicted property values for the $i$-th example. Results reveal a significant gap in performance between the purely supervised model and those benefiting from SNIP's prior knowledge. This performance gap can be attributed to SNIP's pre-trained, semantically rich representations, enabling enhanced generalization to unseen functions. Additionally, fine-tuning the SNIP encoder results in marginal performance gains, indicating the model's capability to adapt to different downstream tasks.

**Qualitative Findings.** To delve deeper into the power of SNIP's representations, we compared its pre-finetuning and post-finetuning latent spaces against that of a supervised model lacking pre-training, using t-distributed Stochastic Neighbor Embedding (t-SNE) (van der Maaten & Hinton, 2008). The visualizations are color-coded by the corresponding properties (Fig. 2). Consistent with the quantitative outcomes, the supervised model's latent space, shown in Fig. 2(a), exhibits limited structural coherence. In contrast, SNIP's latent space in Fig. 2(b) shows pronounced clustering and distinct property trends. Notably, further fine-tuning of the encoder for these prediction tasks, depicted in Fig. 2(c), results in a more structured latent space, marked by clearer linear trends in properties. This finding underscores SNIP's quantitative advantages and its flexibility in adapting to downstream tasks.

**Low Data Regime Analysis.** We evaluated how training sample size influences the test $R^2 = 1 - NMSE$ scores for predicting *NCR*, assessing three model variants on a fixed 1K-sample test set (Fig. 3). In low data regime scenarios with as low as just 100 training samples, the supervised model's score fell sharply to 0.292, while both SNIP variants maintained scores above 0.745. Upon increasing the training sample size to 1M, all models showed improvement; however, SNIP variants continued to lead. We observe that the supervised baseline model might approach SNIP's performance with more training data, which is reasonable, since this model is specialized only for the prediction of this property. However, SNIP's value lies in its flexibility - the pre-trained representations can be efficiently adapted to new tasks. These results emphasize SNIP's superior generalization from limited data, underscoring the SNIP's rich semantic encodings.

## 5 USING SNIP FOR SYMBOLIC REGRESSION

SNIP aims to synergize symbolic and numeric reasoning through mutual learning, offering enhanced capabilities for tasks that require both numeric-symbolic understanding and generation. A paramount task in this context is *Symbolic Regression* (SR), which identifies interpretable symbolic equations to represent observed data. Essentially, SR transforms numeric observations into underlying mathematical expressions, thereby making it a numeric-to-symbolic generation task. The significance of SR extends to revealing functional relations between variables and offers an ideal

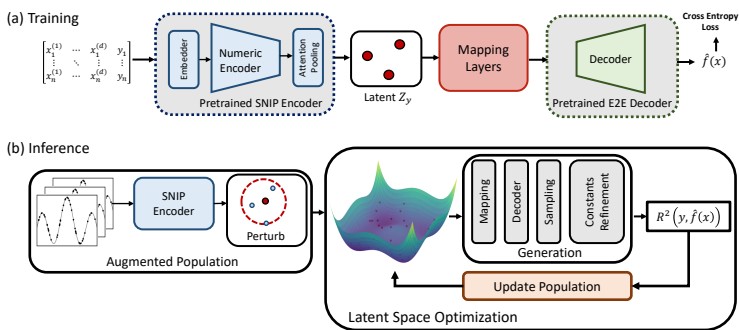

Figure 4: Using SNIP for Symbolic Regression: **(a) Training** includes adding an expression generation module atop SNIP's numeric encoder; **(b) Inference** aims to enhance expressions by optimizing within SNIP's interpolatable latent space.

benchmark for evaluating SNIP's pre-trained numeric representations. Recent advancements in SR leverage encoder-decoder Transformer frameworks (Biggio et al., 2021; Kamienny et al., 2022). Therefore, to effectively undertake SNIP for SR, we perform the following two steps: First, training an expression generation decoder on top of the SNIP's numeric encoder for generating the symbolic functions. Second, conducting latent space optimization (LSO) within SNIP's interpolatable latent space, enriched by pre-training, to further enhance the equation generation.

## 5.1 SR MODEL ARCHITECTURE AND TRAINING

We build the SR model upon SNIP's numeric encoder $\mathcal{E}_\theta^V$ which transforms numeric data into semantically rich embeddings. On top of this encoder, we implement an expression generation module $\mathcal{G}_\omega$ that integrates an expression decoder $\mathcal{D}_\phi$ and a mapping network $g_\gamma$ to generate symbolic expressions: $\mathcal{G}_\omega = \mathcal{D}_\phi \circ g\left(\mathbf{Z}_V; \gamma\right)$.

**Expression Decoder.** To use SNIP for SR, we overlay an expression generation decoder $\mathcal{D}_\phi$, after SNIP's numeric encoder (shown in Fig. 4(a)). This decoder, which utilizes a multi-layer Transformer (Biggio et al., 2021; Kamienny et al., 2022), is trained to map numeric encodings into symbolic expressions, aiming to minimize the divergence between the predicted $\hat{f}$ and actual functions $f$.

**Mapping Network.** Inspired by the ClipCap approach (Mokady et al., 2021) in the field of image captioning, which integrates CLIP's pre-trained image embeddings with GPT-2 pre-trained text generation model through a learnable mapping network, we adopt a similar strategy for SR. As shown in Fig. 4(a), to facilitate integration with the E2E's (Kamienny et al., 2022) pre-trained SR decoder $(\mathcal{D}_\phi^{E2E})$, we introduce a learnable Mapping Network $g_\gamma$. This module translates SNIP's numeric embeddings $\mathbf{Z}_V$ into a compatible input for $\mathcal{D}_\phi^{E2E}$. Specifically, $g : \mathbb{R}^{d_{emb}} \to \mathbb{R}^{M \times d_{emb}}$ reshapes SNIP embeddings into a sequence with maximum length $M$. This approach lets us leverage the existing pre-trained SR decoder without the need for training from scratch.

**Training.** The training objective is to minimize the token-matching cross-entropy loss $\mathcal{L}$ between the predicted $\hat{f}$ and ground-truth $f$ symbolic expressions: $\mathcal{L}(\hat{f}, f) = -\frac{1}{|f|} \sum_j \log P(\hat{t}_j | t_1, \ldots, t_{j-1}; \mathcal{G}_\omega)$, where $P(\hat{t}_j | t_1, \ldots, t_{j-1}; \mathcal{G}_\omega)$ is the conditional probability of the $j$-th token in $\hat{f}$, given the preceding true tokens. Here, the decoder is initialized from pre-trained weights (Kamienny et al., 2022) and trained jointly with the mapping network to learn numeric-to-symbolic expression generation. More details on the model design and training implementation can be found in App. E.

## 5.2 SEMANTIC LATENT INSIGHTS FOR SR

Traditional SR methods rely on searching within the vast equation landscape, dealing with the dual challenges of combinatorial complexity and limited prior knowledge (Burlacu et al., 2020; Schmidt & Lipson, 2009). Recent approaches incorporate deep learning to better navigate this space, integrating learned numeric-to-symbolic priors into the search process (Udrescu & Tegmark, 2020; Petersen et al., 2021; Biggio et al., 2021; Kamienny et al., 2022). Yet, these are also often constrained by their reliance on the function search techniques at the decoding stage (Mundhenk et al., 2021; Holt et al., 2023; Landajuela et al., 2022), perpetuating the limitations. For example, in the Genetic Programming (GP) function search techniques, mutation and breeding steps across 'winning' sub-expressions are prone to significant deviations in a function's numeric behavior. This emphasizes the

necessity for a better search strategy attuned to the semantics of the function. Recently, alternative strategies, like latent space learning of symbolic functions through Variational Autoencoders (VAEs) (Popov et al., 2023; Mežnar et al., 2023), trained exclusively for symbolic function reconstruction, do show promise but fall short by neglecting numeric behaviors essential for SR tasks.

In contrast, SNIP offers a novel solution through a task-agnostic joint learning paradigm. This joint learning approach imprints the latent space with a wealth of integrated symbolic and numeric semantics that serve as a high-dimensional 'semantic fingerprint' for various function behaviors and their inherent similarities. Therefore, unlike the latent space in (Popov et al., 2023; Mežnar et al., 2023), SNIP's task-agnostic latent space embodies a robust numeric-symbolic prior, providing an ideal landscape for SR search. By augmenting SNIP's numeric encoder with an expression generation decoder (as shown in Fig 4), we can create a generative latent space—a crucial asset for the numeric-to-symbolic generation task of SR. Our empirical investigations on the generative latent space further enrich this narrative. The innate ***interpolatability* of SNIP's latent space**, as demonstrated in Fig.5, suggests a meaningful correlation between latent space

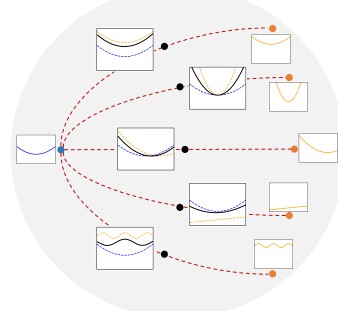

Figure 5: Interpolatability of SNIP numeric latent space.

representations and their corresponding numeric behaviors. In this figure, for a source function $\boldsymbol{Z}_V^s$ (blue curve) and a destination function $\boldsymbol{Z}_V^d$ (orange curves), we linearly interpolate within the numeric encoded vectors to obtain $\boldsymbol{Z}_V^{int}$. This interpolated embedding is decoded into a symbolic function $\hat{f} = \mathcal{G}_\omega(\boldsymbol{Z}_V^{int.})$. Upon computing $\hat{f}$ over dataset $\boldsymbol{x}$, we find that *the interpolated function exhibits a behavior that is semantically in between the source and destination functions.* This is a significant advantage for nuanced search and explorations during the symbolic discovery process. Moreover, the fixed dimension $d_{emb}$ of this space, which is substantially lower than the combinatorial optimization space of equations, streamlines the search process. Given these attributes, SNIP's generative latent space stands as a compelling candidate for a more effective approach to SR.

### 5.3 SNIP LATENT SPACE OPTIMIZATION

As shown in Fig. 5, SNIP latent space interpolation shows a meaningful correlation with the functions' numeric pattern. This observation compels us to undertake a more comprehensive exploration of the latent space. Specifically, to fully harness the expressive capabilities of pre-trained SNIP embeddings in the context of SR, we employ Latent Space Optimization (LSO) as outlined in Fig. 4(b). This optimization process involves a stochastic search over latent space $\boldsymbol{Z}_V$, with the objective of maximizing numerical fitness accuracy. To benefit from both prior knowledge of pre-trained model and capabilities of search method, we initialize the search population by augmenting the given dataset into a partitioned population $\mathcal{P} = \{\mathcal{P}_1, \mathcal{P}_2, \mathcal{P}_3\}$. Specifically, $\mathcal{P}_1$ contains encodings from random sub-samples, with size $n < N$ of the original data; $\mathcal{P}_2$ includes encodings from sampled inputs with their target values $\boldsymbol{y}$ perturbed by random Gaussian noise (perturb and then encode); and $\mathcal{P}_3$ includes perturbed encodings from a fixed sampled data (encode and then perturb). Each agent $p$ with representation $\boldsymbol{Z}_V^p$ is evaluated using a fitness function based on the $R^2$ fitting metric. Candidate symbolic functions are generated for each agent by feeding encodings to the expression generation module $\hat{f}_p = \mathcal{G}_\omega(\boldsymbol{Z}_V^p)$. The functions' constants are then refined using BFGS (Fletcher, 1987), with a goal of optimizing the $R^2$ score against training data (Kamienny et al., 2022). Then, updates to the latent population are carried out using a *gradient-free optimizer*, which accommodates the non-differentiable characteristics of the function generation evaluation metrics. This latent optimization process runs for $T$ iterations or until achieving a predefined $R^2_{stop}$ criterion. The optimal symbolic function $\hat{f}^*$ is then evaluated on a holdout test set. Overall, LSO leverages SNIP's rich latent space to efficiently transform symbolic regression's combinatorial search into continuous optimization of fitting performance. Details on the LSO algorithm and implementation are in App. E. An ablation study analyzing the impact of LSO and choice of optimization algorithm is also provided in App. E.5.

### 5.4 EVALUATION ON SRBENCH

**Datasets.** SNIP was assessed on PMLB datasets (Olson et al., 2017) outlined in SRBench (La Cava et al., 2021), including: 119 *Feynman* equations (Udrescu & Tegmark, 2020), 14 *ODE-Strogatz* challenges (La Cava et al., 2016), and 57 *Black-box* regression tasks without known underlying func-

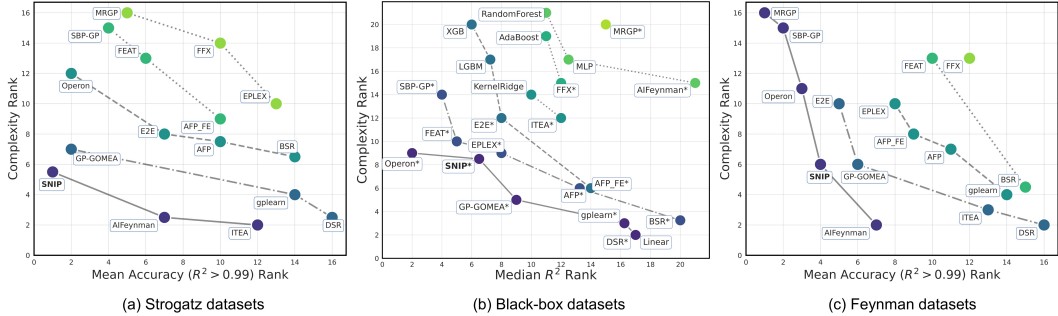

| (a) Strogatz datasets | (b) Black-box datasets | (c) Feynman datasets |

Figure 6: Pareto plots comparing $R^2$ and equation complexity of all methods across **SRBench datasets:** (a) ***Strogatz***, (b) ***Black-box***, and (c) ***Feynman***. Using SNIP for SR yields strong fitting-complexity trade-off, evidenced by its first Pareto-front locating in all datasets. Here, each point depicts a method's median ranking within the data group, with lines/colors signifying Pareto dominance. The "*" marks SR methods in the *Black-box* datasets.

tions. For specifics on each dataset, refer to App. E. Leveraging the E2E's SR decoder (Kamienny et al., 2022) for our decoder initialization, which is trained for $D \leq 10$, we similarly constrained SNIP's pre-training and evaluation to datasets with continuous features and dimensionality $D \leq 10$. Also, since the range of target values $\boldsymbol{y}$ is important, especially for predicting the constants, we do not normalize $\boldsymbol{y}$ for this task. More details on the experiment settings are provided in App. E.

**Results.** Fig. 6 illustrates SNIP's performance against the recent end-to-end (E2E) transformer SR model (Kamienny et al., 2022) and all the SRBench baselines. The Pareto plots exhibit rankings for *Fitting Accuracy* against *Model Complexity*. The model's accuracy is evaluated using $R^2$ and its complexity is evaluated as the number of nodes in the expression tree of the generated equation (La Cava et al., 2021). Here, SNIP shows a strong accuracy-complexity balance, placing on the first Pareto-front across all datasets. On ***Strogatz*** datasets, SNIP demonstrates top-tier accuracy of $0.928$, outperforming all the leading baselines. For ***Black-box*** datasets, SNIP again shows competitive accuracy while achieving lower complexity ($47.52$) than the competitive Operon baseline ($64.95$). On ***Feynman*** datasets, SNIP locates the Pareto frontier, offering better complexity than Operon ($31.63$ vs. $69.87$) and better accuracy than AIFeynman ($0.882$ vs. $0.798$) baselines. More detailed results on the SRBench datasets can be found in App. E.

## 6 DISCUSSION AND CONCLUSION

We introduced SNIP, a multi-modal symbolic-numeric pre-training model that learns how to associate the symbolic and numeric aspects of mathematical functions. We showed that SNIP exhibits remarkable capabilities in estimating cross-modal mathematical properties, particularly in low data regime scenarios, outperforming fully-supervised models. Also, by leveraging the latent space that SNIP constructs—capturing both functional behaviors and symbolic forms—the model demonstrates competitive performance in symbolic regression, even when compared to leading GP baselines. While SNIP showcases robustness and versatility in integrating symbolic and numeric learning, it has notable limitations. It struggles with data patterns that cannot be clearly expressed as closed-form mathematical functions. Also, its performance is tied to the pre-defined data generation protocol, adopted from (Lample & Charton, 2020; Kamienny et al., 2022), which sets constraints on factors such as input dimensionality, and the vocabulary of mathematical operators. For example, the current protocol limits input dimensions to $D \leq 10$ due to the exponential increase in expression complexity at higher dimensions. Exploring higher-dimensional settings is an interesting avenue for future research that would likely require significant updates to the data generation protocol. Despite these limitations, SNIP has a wide range of capabilities, presenting a powerful tool in the intersection of symbolic and numeric mathematics. Future research can focus on potential applications of SNIP, from using numeric guidance in symbolic-to-symbolic tasks such as function integration to using symbolic guidance for numeric-to-numeric tasks such as zero-shot extrapolation and super-resolution. Also, the SNIP's learned representations could serve as a foundation for innovative evaluation metrics of symbolic-numeric proximity, as well as efficient data and feature valuation.

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
