APPENDIX

## A  PRE-TRAINING DATA DETAILS

We provide additional details regarding the pre-training data employed for pre-training SNIP. In our approach, SNIP is pre-trained on a large synthetic dataset of paired numeric and symbolic data, utilizing the data generation technique from (Kamienny et al., 2022). Each example consists of a set of $N$ points $(\boldsymbol{x}, y) \in \mathbb{R}^{D+1}$ and an associated mathematical function $f(\cdot)$, such that $y = f(\boldsymbol{x})$. These examples are generated by first sampling a function $f$, followed by sampling $N$ numeric input points $\boldsymbol{x}_i; i = 1, \ldots, N \in \mathbb{R}^D$ from $f$, and then calculating the target value $y_i = f(\boldsymbol{x}_i)$.

### A.1  SAMPLING OF FUNCTIONS

To generate random functions $f$, we employ the strategy outlined in (Kamienny et al., 2022; Lample & Charton, 2020), building random trees with mathematical operators as nodes and variables/constants as leaves. This process includes:

**Input Dimension Selection.** We begin by selecting the input dimension $D$ for the functions from a uniform distribution $\mathcal{U}(1, D_{max})$. This step ensures variability in the number of input variables.

**Binary Operator Quantity Selection.** Next, we determine the quantity of binary operators $b$ by sampling from $\mathcal{U}(D - 1, D + b_{max})$ and selecting $b$ operators randomly from the set $\mathcal{U}(+, -, \times)$. This step introduces variability in the complexity of the generated functions.

**Tree Construction.** Using the chosen operators and input variables, we construct binary trees, simulating the mathematical function's structure. The construction process is performed following the method proposed in (Kamienny et al., 2022; Lample & Charton, 2020).

**Variable Assignment to Leaf Nodes.** Each leaf node in the binary tree corresponds to a variable, which is sampled from the set of available input variables ($x_d$ for $d = 1, \ldots, D$).

**Unary Operator Insertion.** Additionally, we introduce unary operators by selecting their quantity $u$ from $\mathcal{U}(0, u_{max})$ and randomly inserting them from a predefined set ($\mathcal{O}_u$) of unary operators where $\mathcal{O}_u = [\text{inv}, \text{abs}, \text{pow2}, \text{pow3}, \text{sqrt}, \sin, \cos, \tan, \arctan, \log, \exp]$.

**Affine Transformation.** To further diversify the functions, we apply random affine transformations to each variable ($x_d$) and unary operator ($u$). These transformations involve scaling ($a$) and shifting ($b$) by sampling values from $D_{\text{aff}}$. In other words, we replace $x_d$ with $ax_d + b$ and $u$ with $au + b$, where $(a, b)$ are samples from $D_{\text{aff}}$. This step enhances the variety of functions encountered during pre-training and ensures the model encounters a unique function each time, aiding in mitigating the risk of overfitting as well as memorization.

### A.2  SAMPLING OF DATAPOINTS

Once have generated a sample function $f$, we proceed to generate $N$ input points $x_i \in \mathbb{R}^D$ and calculate their corresponding target value $y_i = f(x_i)$. To maintain data quality and relevance, we follow the guidelines from (Kamienny et al., 2022), which include: *Discarding and Restarting:* If any input point $x_i$ falls outside the function's defined domain or if the target value $y_i$ exceeds $10^{100}$, we discard the sample function and restart the generation process. This ensures that the model learns meaningful and well-behaved functions. *Avoidance and Resampling:* Avoidance and resampling of out-of-distribution $x_i$ values provide additional insights into $f$ as it allows the model to learn its domain. This practice aids the model in handling input variations. *Diverse Input Distributions:* To expose the model to a broad spectrum of input data distributions, we draw input points from a mixture of distributions, such as uniform or Gaussian. These distributions are centered around $k$ randomly chosen centroids, introducing diversity and challenging the model's adaptability.

The generation of input points involves the following steps:

**Cluster and Weight Selection.** We start by sampling the number of clusters $k$ from a uniform distribution $\mathcal{U}(1, k_{max})$. Additionally, we sample $k$ weights $\{w_j \sim \mathcal{U}(0, 1)\}_{j=1}^k$, which are normalized to $\sum_j w_j = 1$.

**Cluster Parameters.** For each cluster, we sample a centroid $\mu_j \sim \mathcal{N}(0,1)^D$, a vector of variances $\sigma_j \sim \mathcal{U}(0,1)^D$, and a distribution shape $D_j$ from $\{\mathcal{N}, \mathcal{U}\}$ (Gaussian or uniform). These parameters define the characteristics of each cluster.

**Input Point Generation.** We sample $[w_j N]$ input points from the distribution $D_j(\mu_j, \sigma_j)$ for each cluster $j$. This sampling with different weights from different distributions ensures the sampling of a diverse set of input points with varying characteristics.

**Normalization.** Finally, all generated input points are concatenated and normalized by subtracting the mean and dividing by the standard deviation along each dimension.

## B PRE-TRAINING IMPLEMENTATION DETAILS

### B.1 MODEL DESIGN DETAILS

**Numeric Encoder.** The numeric encoding mechanism of our SNIP closely follows the design presented by (Kamienny et al., 2022), as highlighted in Sec. 3. Firstly, for each instance in a given batch, the encoder receives $N = 200$ numeric input points, $(\boldsymbol{x}, \boldsymbol{y})$, from a space $\mathbb{R}^{D+1}$. Each of these points is tokenized into a sequence of length $3(D+1)$. An embedding module maps these tokens into a dense representation with an embedding size of $d_{\text{emb}} = 512$. The sequences are then processed in the embedder module by a 2-layer feedforward neural network. This network projects input points to the desired dimension, $d_{\text{emb}}$. The output from the embedder is passed to a Transformer encoder, a multi-layer architecture inspired by (Vaswani et al., 2017). Our specific implementation has 8 layers, utilizes 16 attention heads, and retains an embedding dimension of $512$. A defining characteristic of our task is the permutation invariance across the $N$ input points. To accommodate this, we've adopted the technique from (Kamienny et al., 2022), omitting positional embeddings within the numeric Transformer encoder. In our design, this specialized encoder variant is termed $Enc^V$. The representation generated at the $l$-th layer of the encoder is represented as $\boldsymbol{V}_l$. The process can be summarized as $\boldsymbol{V}_l = Enc_l^V(\boldsymbol{V}_{l-1})$. Here, the index $l$ spans from 1 to $L_V$, where $L_V = 8$ denotes our encoder's total layers. Post encoding, for each instance in the batch, the numeric encoder's sequence outputs, $\boldsymbol{V}_{L_V} \in \mathbb{R}^{N \times d_{\text{emb}}}$, are compressed into a representation for the whole sequence, $\boldsymbol{Z}_V \in \mathbb{R}^{d_{\text{emb}}}$. This representation captures the essence of the entire numeric sequence and is achieved through an attention-pooling mechanism, detailed in Sec. 3.1.

**Symbolic Encoder.** Our SNIP's symbolic encoding component draws inspiration from the model used in (Lample & Charton, 2020), as highlighted in Sec. 3. This encoder is designed to process mathematical symbolic expressions with a maximum length of $200$. These expressions encapsulate the true functional relationships underlying the numeric data fed to the numeric encoder. The expressions are tokenized using a prefix order tree traversal. We employ the vocabulary defined by (Kamienny et al., 2022), crafted to comprehensively represent mathematical equations. It includes symbolic entities like variables and operators, along with numeric constants. Constants are tokenized into three parts, consistent with the tokenization method outlined in Sec. 3.1. Sequence boundaries are indicated with special tokens $[\langle BOS \rangle]$ and $[\langle EOS \rangle]$. Tokens are transformed into dense vectors of dimension $d_{\text{emb}} = 512$ using an embedder module. This module essentially functions as an embedding matrix for the employed vocabulary. To maintain uniform input lengths, sequences are padded to a maximum length of $M = 200$ and then projected to the desired embedding dimension. This dimensionality is aligned with the numeric encoder's. The embedded sequences are processed through a Transformer encoder, characterized by its multi-layer architecture as described by (Vaswani et al., 2017). Similarly, our specific configuration for this encoder consists of 8 layers, utilizes 16 attention heads, and retains an embedding dimension of $512$. Contrary to the numeric encoder, the sequence order in symbolic expressions holds significance. Consequently, we are including positional embeddings into this Transformer encoder variant. We denote this encoder as $Enc^S$, and its layer-wise representations are articulated as $\boldsymbol{S}_l = Enc_l^S(\boldsymbol{S}_{l-1})$, iterating from layer 1 to the maximum layer $L_S = 8$. Similar to the numeric encoder's approach, the symbolic encoder condenses its Transformer outputs $\boldsymbol{S}_{L_S} \in \mathbb{R}^{M \times d_{\text{emb}}}$ for each expression into a compact representation, $\boldsymbol{Z}_S \in \mathbb{R}^{d_{\text{emb}}}$. This aggregation leverages the attention-pooling technique detailed in Sec. 3.2.

### B.2 TRAINING DETAILS

Following the extraction of coarse representations from both symbolic and numeric encoders, our focus shifts to harmonizing the embeddings from these encoders. The aim is to closely align em-

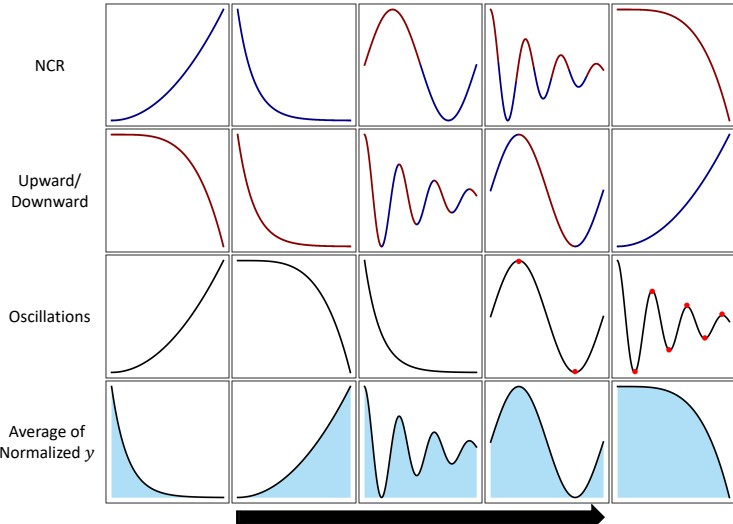

Figure 7: Properties are qualitatively illustrated using five sample functions. Within each row, the plots are arranged according to their respective property values. Colors represent distinct function phases corresponding to the property (e.g., convexity vs. nonconvexity in the first row, upward vs. downward in the second row). Additionally, in the third row, red points highlight instances of change in the y-coordinate.

beddings representing corresponding symbolic-numeric pairs, while ensuring a discernible distance between unrelated pairs. As discussed in Sec. 3.3, this alignment process leverages a symmetric cross-entropy loss calculated over similarity scores, with the specific approach being informed by a contrastive loss mechanism. This ensures effective learning of the correspondence between numeric and symbolic data pairs. Our optimization process is facilitated by the `Adam` optimizer, operating on a batch size of $B = 256$ (symbolic, numeric) data pairs. The learning rate initiation is set at a low $10^{-7}$, which is then gradually warmed up to $4 \times 10^{-5}$ over an initial span of 100K steps. Subsequently, in line with the recommendations of (Vaswani et al., 2017), we apply an inverse square root decay based on the step count to adjust the learning rate. Our model undergoes training for a total of $\approx 220$ epochs, with each epoch comprising $1,000$ steps. This translates to the processing of $256 \times 1\text{K} = 256\text{K}$ (symbolic, numeric) pair samples for each epoch. Given the on-the-fly data generation mechanism, as highlighted in Sec. A, the cumulative volume of data encountered during pre-training approximates a substantial 60M (symbolic, numeric) pair samples. For training, we utilize 4 GPUs, each equipped with 48GB of memory. Given this configuration, the processing time for a single epoch is approximately two hours.

## C   Details of Using SNIP for Cross-Modal Property Prediction

### C.1   Properties Definition

In this section, we define the numeric mathematical properties that we use to evaluate the pre-trained SNIP model. The experiments include understanding and predicting numeric properties, i.e., properties that describe the behavior of numeric dataset, from symbolic forms of functions. The formal definitions of these properties are described in the following paragraphs and Fig. 7 qualitatively illustrates what each of the numeric properties represent.

**Non-Convexity Ratio:**   Non-Convexity Ratio (NCR) is defined to quantify the relative convexity (or non-convexity) of the functions as one of the properties depending on the numeric behavior of the functions. Hence, directly predicting this property from the symbolic form of the function is a complex task. To quantify the non-convexity ratio, we employ Jensen's inequality as a fundamental measure (Tamura & Gallagher, 2019). In our approach, we focus on the one-dimensional equations with numeric dataset $\{\boldsymbol{x}, \boldsymbol{y}\}$. Considering a function $f : \mathcal{D} \to \mathbb{R}$ where $\mathcal{D}$ is a convex subset of $\mathcal{R}$, $f$ is a convex function if $\forall x_1, x_2 \in \mathcal{D}$ and $\forall \lambda \in [0, 1]$:

$$f(\lambda x_1 + (1 - \lambda)x_2) \leq \lambda f(x_1) + (1 - \lambda)f(x_2).$$

We rely on the training datasets with non-regularly sampled points to calculate the approximate NCR. To this end, we perform multiple trials to examine Jensen's inequality criterion. For each trial, we randomly select three data points $\{(x_i, f(x_i)), (x_j, f(x_j)), (x_k, f(x_k))\}$ which are sorted based on $x$ in ascending order. The convexity criterion holds on these points if

$$f(x_j) \leq \frac{(x_k - x_j) \cdot f(x_i) + (x_j - x_i) \cdot f(x_k)}{x_k - x_i} + \epsilon, \tag{3}$$

where $\epsilon$ is a very small number ($\epsilon = 10^{-9}$) to avoid numerical precision errors. Therefore, for trial $t$, we define the success as

$$\xi_t = \begin{cases} 1 & \text{if (3) holds,} \\ 0 & \text{otherwise.} \end{cases}$$

Finally, the non-convexity ratio (NCR) is computed over the total number of trials $T$ as

$$NCR = 1 - \frac{1}{T} \sum_{t=1}^{T} \xi_t.$$

Therefore, if a function is always convex over the range of training data points, `NCR=0`, and if it is always non-convex, it would have `NCR=1`. Functions that have both convex and non-convex sections in the range of $x$ will have `NCR` $\in (0, 1)$.

**Upwardness:** The 'Upward/Downwardness' of a one-dimensional numeric dataset is defined to gauge the proportion of points within the training range where the function exhibits increasing or decreasing behavior. To compute this metric on the sorted dataset $\{\boldsymbol{x_s}, \boldsymbol{f(x_s)}\}$, we examine every consecutive pair of points $\{x_i, x_{i+1}\}$ to determine if they demonstrate an upward or downward trend. We then define $u_i$ as follows:

$$u_i = \begin{cases} 1 & \text{if } f(x_{i+1}) > f(x_i) + \epsilon, \\ -1 & \text{if } f(x_{i+1}) < f(x_i) - \epsilon, \\ 0 & \text{otherwise.} \end{cases}$$

Finally, the upwardness metric `UP` is computed as the average upwardness $\text{UP} = \sum_{i=1}^{N-1} u_i$, where $N$ is the number of points in the dataset. Therefore, if a function is monotonically increasing the range of $x$ in training points, the upwardness measure is 1, and if it is monotonically decreasing, the metric will be $-1$. Functions that have both sections in the range of $x$ will have `UP` $\in (-1, 1)$.

**Oscillation** For this metric, we aim to quantify the degree of oscillatory behavior exhibited by the numeric data. This is approximated by counting the instances where the direction of $y$ changes. Determining the direction of data points follows a similar process to that of the upwardness metric for each consecutive pair. Thus, we tally the occurrences of direction changes while traversing the sorted dataset. Due to the potential variation in the number of changes, we opt for a logarithmic scale to color the plots.

**Average of Normalized** $y$ The overall behavior of the numeric data points $\{\boldsymbol{x}, \boldsymbol{y}\}$ are better represented when the values of $y$ are scaled to a fixed range (here $(0, 1)$), giving $\{\boldsymbol{x}, \boldsymbol{Y}\}$. The average of the normalized values, $\bar{Y}$ can be a measure to distinguish different numeric behaviors, and it can roughly approximate the numerical integral of the normalized function in the defined range of training $\boldsymbol{x}$.

## C.2 ADDITIONAL QUANTITATIVE RESULTS OF CROSS-MODAL PROPERTY PREDICTION

**Evaluation Metrics Overview.** We continue to use **NMSE** as our primary regression metric, providing a standard comparison across different model variants for each cross-modal property prediction task. We also report results for the **Accuracy within Tolerance** ($Acc_\tau$) evaluation metric, reflecting how closely the predicted values align with the true values, within a specified tolerance level. To this end, we first normalize the true and predicted values for each property in the range of $(0, 1)$ based on the range of true values. Subsequently, we calculate the accuracy over $N_{test} = 1000$ test examples as $Acc_\tau = \frac{1}{N_{test}} \sum_{i=1}^{N_{test}} \mathbb{1}\{|\hat{p}_i - p_i| \leq \tau\}$, where $p_i$ and $\hat{p}_i$ are the normalized true and predicted values of the property for the $i$-th example, respectively. Here, we consider an absolute tolerance $\tau = 0.1$.

Table 2: Full results of cross-modal property prediction on four properties showcase SNIP's superiority over the supervised baseline.

| Model | Non-Convexity Ratio | | Upwardness | | Normalized Average $y$ | | Log Oscillations | |
|---|---|---|---|---|---|---|---|---|
| | $\downarrow$ NMSE | $\uparrow Acc_{0.1}$ | $\downarrow$ NMSE | $\uparrow Acc_{0.1}$ | $\downarrow$ NMSE | $\uparrow Acc_{0.1}$ | $\downarrow$ NMSE | $\uparrow Acc_{0.1}$ |
| Base | 1.0000 | 2.4% | 1.0000 | 22.4% | 1.0000 | 52.4% | 1.0000 | 23.1% |
| Supervised | 0.5299 | 56.5% | 0.5356 | 56.3% | 1.0406 | 49.3% | 0.3079 | 75.2% |
| SNIP (frozen) | 0.0731 | 86.1% | 0.0540 | 84.7% | 0.4532 | 64.5% | 0.0683 | 92.6% |
| SNIP (finetuned) | **0.0683** | **92.1%** | **0.0400** | **90.1%** | **0.4074** | **67.7%** | **0.0581** | **92.6%** |

**Chance Level Baselines.**  For each metric, we establish a baseline or chance level to set a comparative standard to better show task difficulty. For NMSE, the chance level is at $NMSE = 1$, representing a prediction that averages the property values without considering the input. The chance level baseline for $Acc_{0.1}$ is calculated based on the assumption that all predictions are equal to the mean (average) property value, $\hat{p}_i = \bar{p}$.

**Detailed Results.**  To offer a more detailed and nuanced perspective on the performance of SNIP for cross-modal property prediction, we delve into its performance across four fundamental properties: Non-Convexity Ratio (NCR), Upwardness, Average of $y$, and Oscillations. Table 2 showcases a thorough comparison of the results from different model variants on these specified mathematical properties. A key aspect of these experiments is the exclusive use of symbolic equations as input for all models, aligning with the cross-modal essence of the tasks. The numeric properties are then predicted for these symbolic inputs, demonstrating the models' ability to bridge symbolic and numeric domains. For consistency and fairness in our evaluation, all models were trained on uniform datasets, each consisting of 10K equations, and then assessed using a separate set of 1K equations for evaluation. These datasets were constructed following the methodology outlined in Sec. A. It's imperative to highlight that, in the context of cross-modal property prediction, SNIP operates with the same quantity of labeled examples as the supervised baselines. However, a critical distinction lies in SNIP's pre-training phase, where it was not exposed to any labeled data. Instead, it engaged in an unsupervised learning process, focusing on deciphering mutual symbolic-numeric similarities in representations. The results presented in Table 2 demonstrate that SNIP, both in its original 'frozen' state and when finetuned, consistently surpasses the performance of supervised models across all evaluated properties. This superiority is evident in both metrics – $NMSE$ and $Acc_{0.1}$. The variation in chance levels across the different properties is particularly revealing, as it highlights the unique challenges inherent to each property. This variance underscores the adaptability and robustness of the SNIP model in navigating and excelling in the diverse landscape of cross-modal property prediction tasks.

## C.3 Additional Qualitative Findings of Cross-modal Property Prediction

In addition to numerical results, we include visual representations of the model's latent features for each property. These visualizations offer a qualitative perspective on how our model captures and represents the underlying characteristics of different properties. Fig. 8 shows a qualitative comparison of pre-finetuning and post-finetuning latent spaces of SNIP against that of supervised task prediction models, using 2-dimensional t-SNE visualizations of the encoded representations. The first two rows (NCR and Upwardness) are replicated from the main body (Fig. 2) for ease of comparison. In each task (row), the plots are colored by the values of the corresponding property. In each task, a training dataset with 10K samples was used to train the model.

The observations from Fig. 8 show that the latent spaces of supervised models (without pre-trained SNIP) are very weakly structured and barely exhibit a recognizable trend for the properties. On the other hand, when the pre-trained SNIP is used, the latent spaces are shaped by the symbolic-numeric similarities of the functions such that numeric properties can be clustered and/or show visible trends in the symbolic encoded representation space $\boldsymbol{Z}_S$. Furthermore, fine-tuning the encoder, as shown in Fig. 8(c), leads to more organized latent spaces with distinct linear property trends.

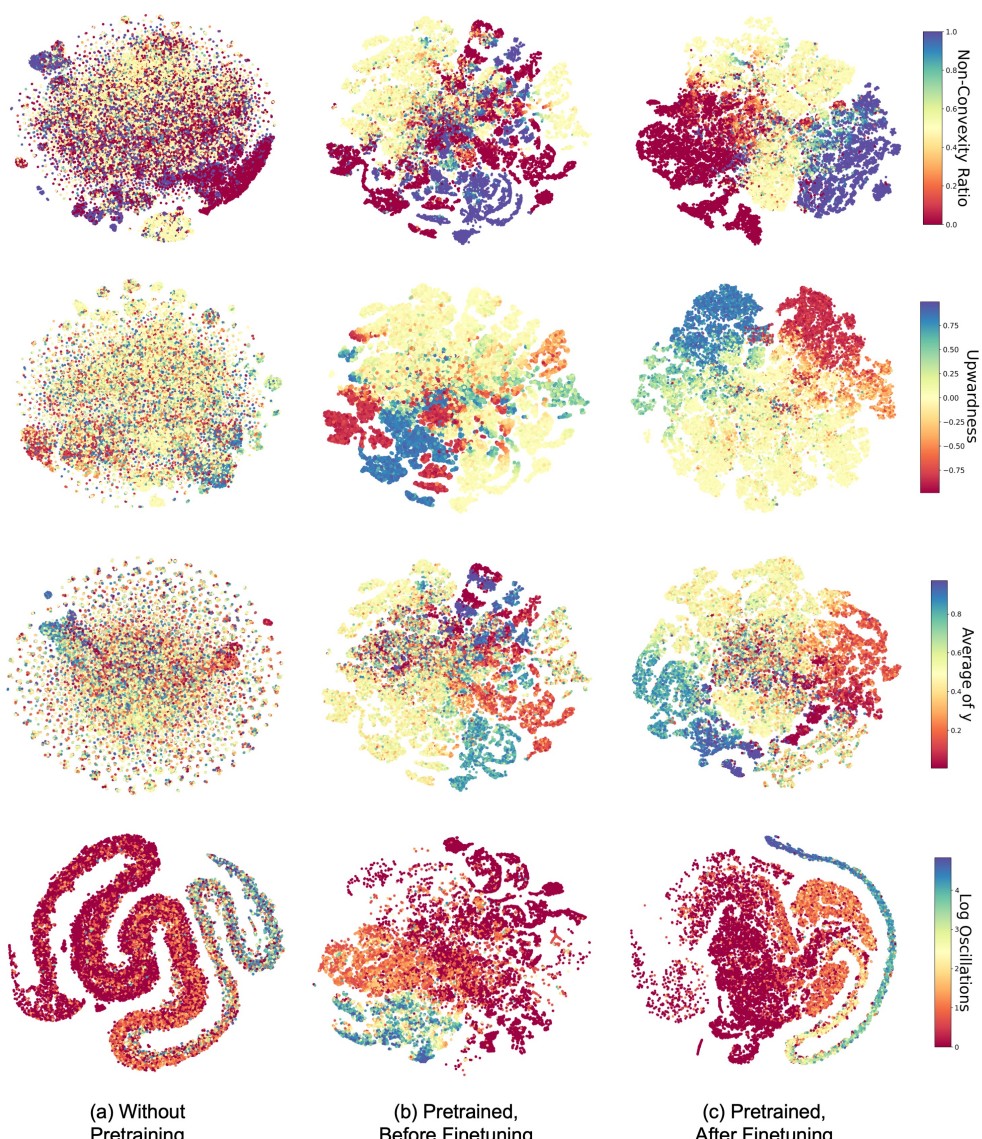

(a) Without
Pretraining

(b) Pretrained,
Before Finetuning

(c) Pretrained,
After Finetuning

Figure 8: 2D t-SNE plots of the symbolic encoded representations for the tasks of predicting numeric mathematical properties: Non-Convexity Ratio, Function Upwardness, Average of $y$, and Oscillations. The plots compare the **(a)** supervised models without pre-training, **(b)** frozen pre-trained SNIP encoder, and **(c)** fine-tuned SNIP encoders for each task.

# D    ADDITIONAL VISUALIZATIONS OF SNIP PRE-TRAINED LATENT SPACE

**Numeric Encoded Representations.**    We show that akin to how symbolic encoded representations are shaped by numeric behaviors, the numeric encoded vectors $Z_V$ are likewise influenced by the symbolic attributes of the corresponding governing equations. To illustrate this, Fig. 9 showcases 2D t-SNE visualizations depicting the learned latent space of SNIP's numeric encoded vectors, color-coded by function (a) complexity and (b) an arbitrarily defined categorization of the functions based on their dominant operators. Further details regarding these two symbolic features are provided below:

*Function Complexity:* Function complexity, as defined in Symbolic Regression (SR) tasks, pertains to the length of the function expressed in prefix order notation,i.e., the number of nodes in the expression tree. Intuitively, functions with a greater number of operators and variables (resulting in longer equations) are considered more complex, often exhibiting correspondingly complex behaviors.

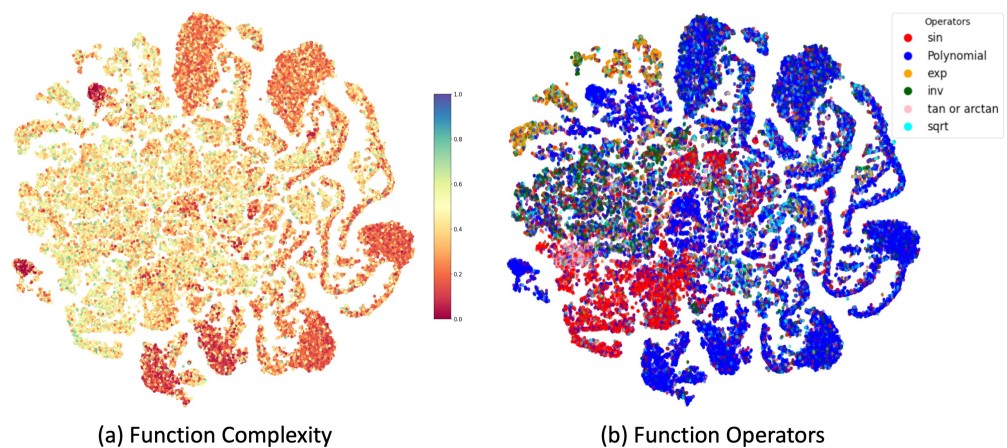

(a) Function Complexity                           (b) Function Operators

Figure 9: 2D t-SNE plots of the pretrained SNIP numeric encoded representations ($\boldsymbol{Z}_V$) colored by **(a)** Function Complexity, and **(b)** Function Classes based on Operators.

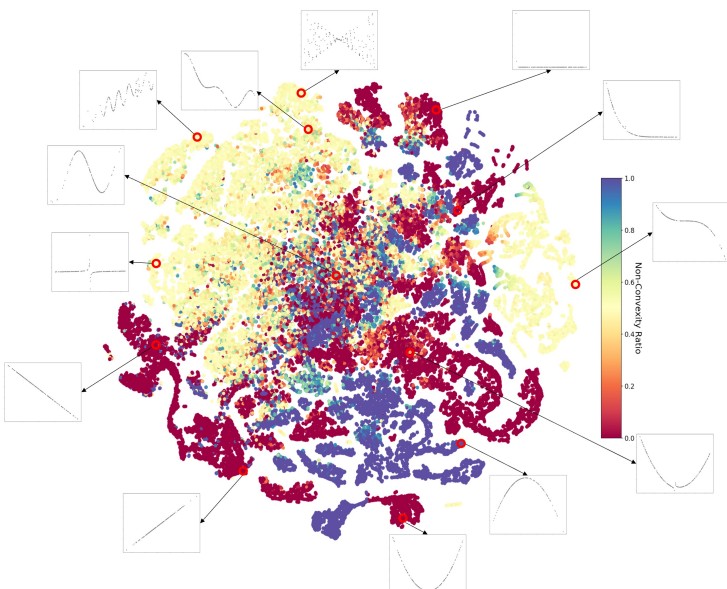

Figure 10: 2D t-SNE plot of the pretrained SNIP symbolic encoded representations ($\boldsymbol{Z}_S$) colored by Non-Convexity Ratio property. Adjacent to the corresponding locations of points in the latent space, the numeric behaviors of selected sample equations are displayed, illustrating the interplay between their symbolic forms and numeric properties. This visualization underscores how both the symbolic and numeric characteristics of functions influence their representation in SNIP's latent space.

*Function Operator Classes:* Mathematical functions can be broadly classified into different classes based on the operators utilized in their expressions, which in turn influence the behavior of the data they describe. It is important to note that a single function may incorporate multiple operators, contributing to the overall complexity of the data's behavior. Additionally, certain operators within a function may hold more significance than others, exerting greater influence on the range and pattern of the data. To categorize the functions, we employ the following guidelines:

First, we consider a prioritized set of unary operators: $\mathcal{O} = \{\mathrm{arctan, tan, exp, sqrt, inv, cos, sin,}$ $\mathrm{pow3, pow2}\}$. If a function exclusively employs one of these operators, it is categorized accordingly. For simplicity, we designate both $\mathrm{pow2}$ and $\mathrm{pow3}$ as `Polynomial`, and we employ `sin` for both $\mathrm{sin}$ and $\mathrm{cos}$. In the event that a function incorporates more than one operator, it is assigned to the category corresponding to the operator of higher priority. It is worth noting that this categorization

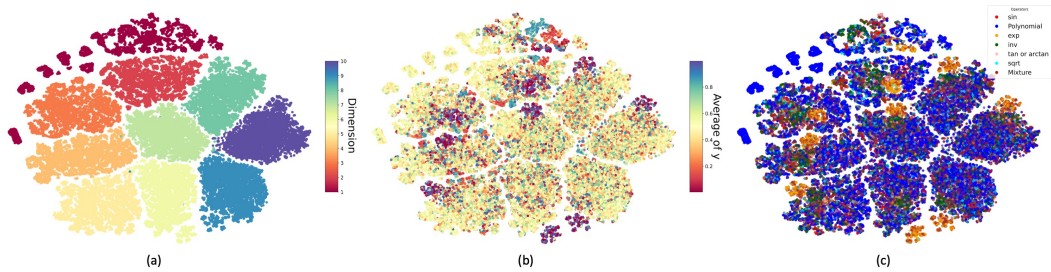

Figure 11: 2D t-SNE plots of the SNIP symbolic encoded representations on up to 10-dimensional datasets, colored by **(a)** dimension of the functions, **(b)** Average of normalized $y$, and **(c)** classes of functions based on their operators.

may not always perfectly capture the behavior of functions, as an operator with lower priority may potentially exert a more dominant influence than another prioritized operator.

**Annotated Latent Space.** To have a closer look to the latent space representation, we also analyze several functions with their position in the learned latent space t-SNE visualization. Fig. 10 shows the same t-SNE plot of $\boldsymbol{Z}_S$ (from the symbolic encoder) colored by NCR property and annotated by the numeric behavior (scaled $y$) of some samples. We can observe that the latent space is shaped by both symbolic input $f(\cdot)$ and numeric data, such that closer points have more similar symbolic and numeric features.

**10-Dimensional SNIP Latent Space Analysis.** Fig. 11 shows the latent space representation of the pre-trained SNIP with numeric datasets of up to 10 dimensions, which is used for the symbolic regression task (so that we can evaluate on SRBecnh and compare with SOTA baselines). We observe that the model can cluster the functions with different dimensions, and within each cluster, it is shaped by the symbolic-numeric similarity of the functions.

# E    DETAILS OF USING SNIP FOR SYMBOLIC REGRESSION

## E.1    IMPLEMENTATION DETAILS

In this section, we provide the details of the model and training procedure for the symbolic regression task. As illustrated in Fig. 4 of the main body, the training step includes learning a mapping module and fine-tuning an expression generation decoder which is borrowed from (Kamienny et al., 2022). We elaborate upon each of the modules and the details of training.

**Expression Decoder.** The pre-trained expression decoder from (Kamienny et al., 2022) is a seq2seq transformer decoder (Vaswani et al., 2017) with 16 attention heads and the same embedding dimensionality of 512. The decoder has 16 layers (deeper compared to the encoders) to enhance its generation capacity.

**Mapping Network.** The learnable Mapping Network $g_\gamma$ translates SNIP's numeric embeddings $\boldsymbol{Z}_V$ into a compatible input for the decoder $\mathcal{D}_\phi^{E2E}$. Therefore, we can use the power of both pre-trained encoder and decoder modules by learning a mapping between these two modules. In fact, $g : \mathbb{R}^{d_{\text{emb}}} \to \mathbb{R}^{L \times d_{\text{emb}}}$ reshapes SNIP embeddings into a sequence with maximum length $L$. To do so, we use a simple Multi-Layer Perceptron (MLP) design with two linear layers. The first layer applies a linear mapping from $\mathbb{R}^{d_{\text{emb}}}$ to $\mathbb{R}^{L d_{\text{emb}}}$, followed by a ReLU activation. This output is reshaped to add the sequence dimension $\mathbb{R}^{L \times d_{\text{emb}}}$, and then passed to the second layer, which applies a linear mapping from $\mathbb{R}^{d_{\text{emb}}}$ to $\mathbb{R}^{d_{\text{emb}}}$. Consequently, the final output retains the shape $\mathbb{R}^{L \times d_{\text{emb}}}$.

**Training** Similar to the suggestions of (Mokady et al., 2021), we found that to effectively learn the simple MLP mapping network, we can let the decoder network to be simultaneously fine-tuned. In this way, the mapping training is less challenging since we have a control over both networks. We train the whole model in two stages. In the first stage, we freeze the SNIP encoder's parameters and only update the mapping network and decoder's parameters. This allows the model to learn the mapping from the fixed encoded representations of numeric datasets to their corresponding symbolic functions. Similar to the pre-training procedure, an Adam optimizer with learning rate warm-up followed by a inverse square root decay based on number of steps is used to train the model with

cross-entropy loss. In the second stage, to enhance the model's generation capacity, we fine-tune the SNIP's encoder along with the other modules. This helps the model to distinguish between the overlapped representations in the encoder, which were not originally trained for the expression generation objective. It also maintains their relative positions obtained from the contrastive loss. In both stages, we use batch size $B = 128$ for training.

### E.2 SNIP LATENT SPACE OPTIMIZATION DETAILS

In this section, we provide the details of the Latent Space Optimization (LSO) on SNIP's encoded representations. This method combines three main advantages that make it suitable for the symbolic regression task.

- By training an expression decoder on top of SNIP encoder, we learn a prior for function generation given the numeric dataset, which is the main advantage of neural symbolic regression models over traditional search methods.
- While neural SR models are trained using token-matching objectives, LSO utilizes a powerful search with the objective of fitting accuracy. Therefore, it can also enjoy the main advantage of the search methods over the pre-trained equation generation methods.
- The most important advantage of this method is that it **exploits the well-organized latent space of SNIP to perform the optimization in a continuous, low-dimensional, and interpolatable latent space** which provides it with a huge benefit over traditional GP functions search techniques.

Algorithm 1 sketches the main steps of LSO. The red lines indicate when the modules of pre-trained model are called, and blue lines indicate when other functions are called.

---

**Algorithm 1:** Latent Space Optimization (LSO) on SNIP Pre-trained Encodings

**Input:** Dataset $\{\boldsymbol{x}, \boldsymbol{y}\}$, sampling size $b$, stopping $R^2_{stop}$, Maximum Iterations $T$

1. Population Generation
**Generate the search population by following the steps below:**

- Generating $p_1 < P$ points by randomly sampling subsets of the original dataset and Encoding $\boldsymbol{Z}_V^i$ for $i \in \{1, \ldots, p_1\}$.
- Generating $p_2 < P$ points by injecting Gaussian noise to the input data points and Encoding $\boldsymbol{Z}_V^i$ for $i \in \{1, \ldots, p_2\}$.
- Generating $p_3 < P$ points by first Encoding a fixed input dataset and then injecting Gaussian noise to the encoded representation $\boldsymbol{Z}_V$ to get $\boldsymbol{Z}_V^i$ for $i \in \{1, \ldots, p_3\}$.
- Combining these points to have a population with size $P = p_1 + p_2 + p_3$.

**for** $T$ *iterations* **do**
    2. Fitness Evaluation
    **for** *each population agent* $\boldsymbol{Z}_V^i$, $i \in \{1, \ldots, P\}$ **do**
        Compute the fitness value following steps below:
        1. Decode the encoded representation with sampling size $b$ to get $\{\tilde{f}_1^i, ..., \tilde{f}_b^i\}$.
        2. Remove functions with duplicate skeletons.
        3. Refine the constants of the remaining functions on the training set using BFGS.
        4. Compute $R^2$ score for each function; Store the highest score as the fitness value of the agent at current iteration $\mathcal{F}_t^i$.
    **end**
    Set $\mathcal{F}_t^* = \max_i(\mathcal{F}_t^i)$ as the best current score.
    **if** $\mathcal{F}_t^* > R^2_{stop}$ **then**
        Return the best function $f^*(\cdot)$ to be evaluated on the testing dataset.
    **end**
    3. Optimization Step
    Call the Gradient-Free Optimizer update rule with $(\boldsymbol{Z}_V, \mathcal{F}_t)$ to get the updated population $\boldsymbol{Z}_V$.
**end**
Report the best function $f^*(\cdot)$, and evaluate it on the testing dataset.

---

Some of the details of these steps are as follows:

**Population Generation.** To combine the use of prior knowledge with the search method, instead of generating random agents in the latent space, we initialize the population by augmenting the given dataset. In algorithm 1, $p_1$, $p_2$, and $p_3$ are selected to be 15, 10, and 25, respectively, summing

up to $P = 50$ to maintain a balance on the performance and the computation time. Each of the augmentations provides a different perspective that we elaborate upon:

- In the first augmentation, $\mathcal{P}_1$, each augmented agent $\boldsymbol{Z}_V^i$ is obtained by first uniformly sampling a subset, with size $n < N$ of the original dataset $\left(\boldsymbol{x}^{\text{sub}_i}, \boldsymbol{y}^{\text{sub}_i}\right) \subseteq (\boldsymbol{x}, \boldsymbol{y})$. Since the maximum sequence length is 200, we set $n = 200$ if $N > 400$, and set $n = \lfloor N/2 \rfloor$ if $N < 400$. Subsequently, we encode the sampled data to get $\boldsymbol{Z}_V^i = \mathcal{E}_\theta^V \left(\left\{\left(\boldsymbol{x}^{\text{sub}_i}, \boldsymbol{y}^{\text{sub}_i}\right)\right\}\right)$.

- For the second augmentation, $\mathcal{P}_2$, each augmented agent $Z_V^i$ is obtained by first perturbing the target values with random Gaussian noise $(\boldsymbol{x}, \boldsymbol{y} + \boldsymbol{\epsilon}_i)$, where $\boldsymbol{\epsilon}_i \sim \mathcal{N}(0, \sigma_i^2 I_n)$, and $\sigma_i \propto i$ to cover different ranges of perturbations for a more diverse search population. Subsequently, we encode the perturbed data to get $\boldsymbol{Z}_V^i = \mathcal{E}_\theta^V \left(\left\{(\boldsymbol{x}, \boldsymbol{y} + \epsilon_i)\right\}\right)$.

- For the third augmentation, $\mathcal{P}_3$, each augmented agent $Z_V^i$ is obtained by first encoding the dataset to get $\boldsymbol{Z}_V = \mathcal{E}_\theta^V \left(\left\{(\boldsymbol{x}, \boldsymbol{y})\right\}\right)$, and then perturbing the encoded vectors using random Gaussian noise. $\boldsymbol{Z}_V^i = \boldsymbol{Z}_V + \mathcal{N}(0, \sigma_i^2 I_{d_{\text{emb}}})$, where $\sigma_i$ varies randomly to achieve a more diverse search population.

**Fitness Evaluation.** To evaluate the fitness of the population $\mathcal{P}$ at iteration $t$, we utilize the expression generation modules with sampling (Fan et al., 2018) to generate $b = 2$ candidates for each agent $\boldsymbol{Z}_V^i$. Following this, candidates with duplicate skeletons are eliminated, and the remaining candidate skeletons undergo a refinement process. In order to refine the constant values, a procedure following (Kamienny et al., 2022) is employed. Specifically, the generated constants (model predictions) serve as initial points, and these constants are further optimized using the BFGS algorithm (Fletcher, 1987). Subsequently, we calculate the $R^2$ score on the training data points, which serves as the fitness values for the population.

**Optimization.** Computing the fitness measure $R^2$ from the generated equation $\tilde{f}_i$ is not a differentiable process. Consequently, we resort to utilizing gradient-free optimization algorithms, which operate without the need for gradient information to update the search population. In this context, swarm intelligence algorithms have proven to be both computationally efficient and effective for continuous spaces. Therefore, we opt for a recently developed swarm algorithm known as the Grey Wolf Optimizer (GWO) (Mirjalili et al., 2014) for updating the population vectors. The GWO algorithm employs a balanced exploration-exploitation strategy based on the current elite population agents, i.e., those agents exhibiting the best fitness values. In this work, we select the maximum iteration $T = 80$, and we use early stopping criterion $R^2_{\text{stop}} = 0.99$. Also, at each iteration, we establish lower and upper bounds for agent positions based on the minimum and maximum values of $Z_V$ across both dimensions and all agents.

### E.3 SRBENCH EVALUATION DATASET DETAILS

In our evaluation of SNIP, we resort to the widely-recognized SRBench, a benchmark known for its challenging and diverse datasets in Symbolic Regression (La Cava et al., 2021). This benchmark aggregates datasets from three primary groups: *Feynman*, *Strogatz*, and *Black-box regression*. A visual representation of these datasets is presented in Fig. 12, illustrating the distribution across groups in terms of dataset count, input dimensions, and the number of datapoints. More details on each of these data groups are given below.

*Feynman*[2]: The Feynman dataset is a significant component of the broader landscape of symbolic regression datasets, with its roots traced back to the renowned *Feynman Lectures on Physics database* series (Udrescu & Tegmark, 2020). The dataset aggregates a collection of 119 distinct equations, as visualized in Fig. 12(a). These equations encapsulate a wide range of physical phenomena, serving as a testament to Feynman's contributions to the realm of physics. The regression input points $(x, y)$ for these equations are meticulously indexed within the Penn Machine Learning Benchmark (PMLB) (La Cava et al., 2021; Olson et al., 2017). The SRBench has further shed light on these equations, adopting them as standards in the evaluation of symbolic regression methodologies. One of the critical constraints of this dataset is the input dimensionality, which has been capped at $D \leq 10$, as depicted in Fig. 12(b). This limit ensures a consistent evaluation scale across multiple symbolic regression challenges. Moreover, an advantage that researchers have with this dataset is the availability of the true underlying functions, eliminating the ambiguity often present in black-box

---

[2]https://space.mit.edu/home/tegmark/aifeynman.html

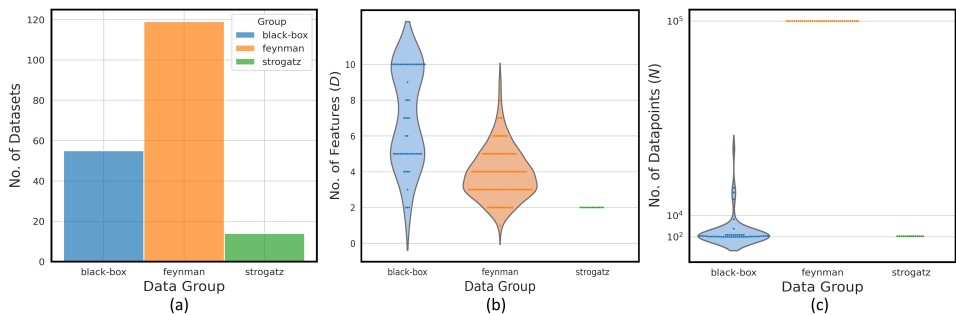

Figure 12: Distribution of datasets across the SRBench *Feynman*, *Strogatz*, and *Black-box* groups: (a) Count of datasets, (b) Spread of input dimensions, and (c) Number of datapoints per dataset.

datasets. Cumulatively, the dataset boasts an impressive count of $10^5$ datapoints, as highlighted in Fig. 12(c).

***Strogatz***[3]**:** The Strogatz dataset is a collection of symbolic regression challenges, drawing inspiration from the realm of nonlinear dynamical systems. Its inclusion in the broader context of symbolic regression evaluations offers a unique perspective, focusing on the intricacies of dynamical behaviors. At the heart of this dataset are 14 distinctive symbolic regression problems, extracted from the esteemed *ODE-Strogatz database* (La Cava et al., 2016). Each of these problems has been crafted to reflect the nuances of nonlinear dynamical systems, underscoring the rich tapestry of dynamical phenomena. The regression input points $(x, y)$ for these challenges are accessible from the Penn Machine Learning Benchmark (PMLB) (Olson et al., 2017). The SRBench has also leveraged these problems, incorporating them into an evaluation framework for symbolic regression (La Cava et al., 2021). An inherent attribute of this dataset is the limitation on input dimensionality, fixed at $D = 2$ (as shown in Fig. 12(b)). This means that for each problem, there are two primary input variables. This restriction facilitates a concentrated exploration of two-dimensional dynamical systems. The true functions, which underlie and generate the data, are also available. Each problem in the Strogatz collection features a dataset containing $N = 400$ data points.

***Black-box***[4]**:** The Black-box dataset group stands as a testament to the versatility and applicability of symbolic regression to real-world complex datasets without known underlying functions, offering challenges that are both diverse in nature and crucial for machine learning evaluations. Primarily sourced from the comprehensive PMLB repository (Olson et al., 2017), the Black-box datasets have garnered significant attention in the SRBench (La Cava et al., 2021), serving as key benchmarks against various state-of-the-art ML regression methods. The primary objective of utilizing the Black-box datasets for symbolic regression is not just about achieving fitting accuracy, but also deciphering models that are interpretable yet better-fitting (compared to ML models), offering insight into the underlying data processes. Ensuring compatibility with the latest methodologies, we constrain our datasets to possess continuous features and an input dimension that doesn't exceed 10: $D \leq 10$. This decision aligns with the training preconditions of the transformer-based SNIP numeric encoder and E2E's SR decoder (Kamienny et al., 2022), which are tailored for an upper limit of $d_{max} = 10$. As a result, out of a broader set, 57 black-box datasets meet the constraint. Datasets of this group offer a mosaic of challenges, stemming from both the real-world and synthetically generated scenarios. An inherent characteristic of these datasets is the noise, varying in levels, which mimics real-world data inconsistencies and imperfections, enhancing the robustness of evaluations. The Black-box collection contains an impressive diversity in terms of number of data points per dataset, ranging from as few as 47 to around 40K. To visually represent this, Fig. 12(c) demonstrates the distribution of datasets across number of datapoints ($N$), with an observable average data point scale around $10^2$ for this group.

---

[3]https://github.com/lacava/ode-strogatz
[4]https://github.com/EpistasisLab/pmlb/tree/master/datasets

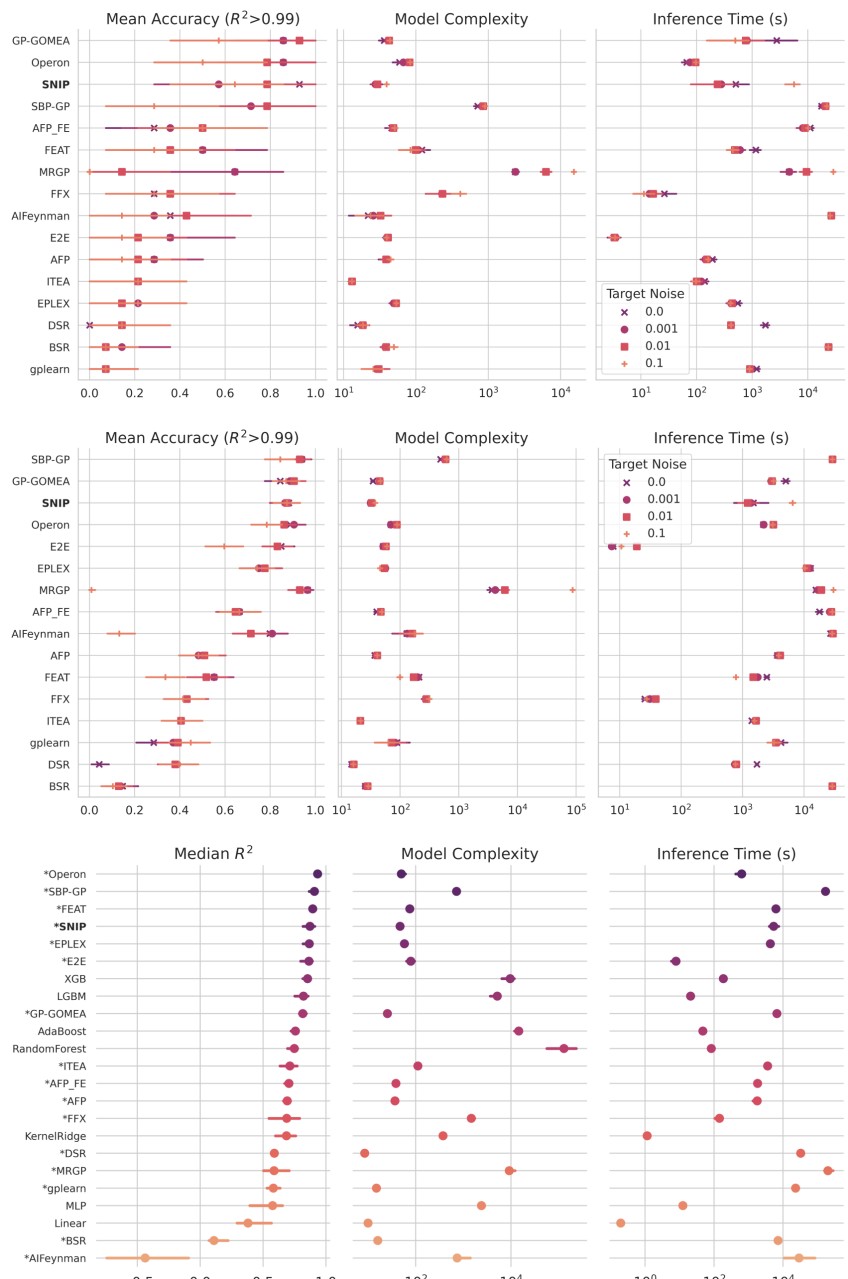

Figure 13: Performance comparison of SNIP and SRBench algorithms in terms of Accuracy-Complexity-Time on **Strogatz** (**top**), **Feynman** (**middle**), and **Black-box** (**bottom**) datasets. For *Feynman* and *Strogatz* dataset, algorithms are sorted based on mean accuracy defined as the ratio of solutions with $R^2 > 0.99$ on test set under various noise levels, and for *Black-box* datasets, the algorithms are sorted based on the median $R^2$ score on test set. SNIP demonstrates a strong balance of performance with relatively low model complexity and competitive inference time compared to GP-based algorithms. The error bars represent the 95% confidence interval and "∗" refers to SR methods for *Black-box* dataset.

## E.4 ADDITIONAL DETAILS FOR SRBENCH EVALUATION EXPERIMENTS

### E.4.1 EXPERIMENT SETTINGS

Aligning with the SRBench evaluations and the methodology from (Kamienny et al., 2022), we partition the observation points of each equation in the SRBench datasets (comprising *Feynman*, *Strogatz*, and *Black-box*) into training and testing subsets with a 75%/25% split.

### E.4.2 RESULTS

**Strogatz:** In Fig. 13 (**top**), we compare the performance of SNIP with the SRBench algorithms on the *Strogatz* dataset. As described in Sec. E.3, the *Strogatz* dataset includes 14 equations from a two-state system governed by a first-order ODE. A key observation is that the end-to-end (E2E) transformer SR model underperforms on this dataset compared to other GP-based models. This underperformance can be attributed to the distinct time-ordered distribution of observations in the *Strogatz* dataset, which deviates considerably from the E2E model's pre-training data. Interestingly, SNIP, despite not being trained on time-ordered data, significantly outperforms not only the E2E transformer SR model but also many other leading SR baselines. In terms of accuracy, SNIP ranks within the top three baselines, specifically when evaluating the proportion of solutions with an $R^2 > 0.99$ across varying target noise levels. Moreover, its inference time is competitive with leading baselines such as GP-GOMEA and Operon. Crucially, SNIP shines in terms of model complexity. It produces expressions that fit well but with lower complexity than top-ranked competitors in Fig.13.

This advantage in balancing accuracy and complexity is also evident in Fig.6(a), where SNIP is positioned on the first Pareto-front, while competitors like GP-GOMEA and Operon fall on the second and third, respectively. This suggests that SNIP offers a superior Accuracy-Complexity trade-off for noise-free data ($\gamma = 0$). Fig. 14(a) further underscores this point by illustrating the Pareto performance of leading SR baselines on the *Strogatz* dataset across various noise levels. As expected, all methods experience a performance drop as target noise increases. Yet, even amidst noise, SNIP consistently maintains its advantageous position in the upper-left corner, indicating its ability to generate expressions that excel both in accuracy and complexity.

**Feynman:** In Fig. 13 (**middle**), we present a comparative analysis of SNIP against the SRBench algorithms on the *Feynman* dataset. As outlined in Sec. E.3, the *Feynman* dataset encompasses 119 unique Feynman equations, representing a broad spectrum of physical phenomena. This figure delineates the positioning of each algorithm in terms of Accuracy-Complexity-Time. Notably, the E2E transformer model exhibits enhanced performance on the *Feynman* dataset relative to the Strogatz dataset, securing a fifth rank in accuracy. SNIP, however, surpasses the performance of not only the E2E transformer SR model but also many top GP baselines. When focusing on accuracy,

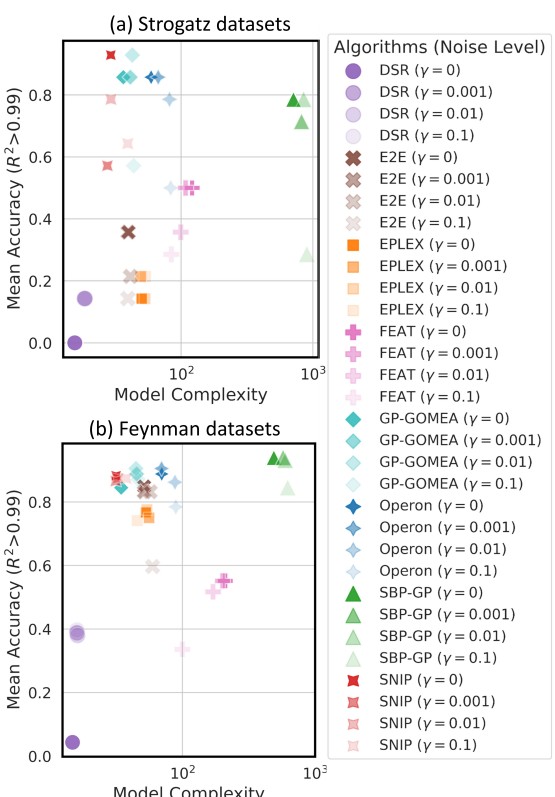

Figure 14: Pareto analysis on (**a**) *Strogatz* and (**b**) *Feynman* datasets, contrasting methods based on fitting accuracy, proportion of solutions with $R^2 > 0.99$, and equation complexity across different noise levels. In both datasets, SNIP mostly stays in the desired upper-left corner, showcasing its robustness in balancing fitting accuracy and complexity even when noise is introduced.

SNIP clinches a spot among the top three baselines, especially when considering solutions boasting an $R^2 > 0.99$ across diverse target noise levels. SNIP often outperforms other competing baselines like GP-GOMEA, SBP-GP, and Operon in terms of inference time. From a complexity perspective, SNIP exhibits superior results against SBP-GP and Operon and demonstrates comparable performance with the GP-GOMEA baseline. This optimal balance between accuracy and complexity was also illustrated in Fig.6(c), showcasing ranking Pareto plots for both metrics. Here, SNIP commands a position on the first Pareto-front, closely approximating the lower-left corner — an indicator of its better ranking in both accuracy and complexity. Among competitors, GP-GOMEA is positioned

on the secondary Pareto level, denoting an inferior Accuracy-Complexity balance relative to SNIP. While Operon and SBP-GP are also placed on the first Pareto-dominance, their placement drifts further to the upper-left corner than SNIP — suggesting the expressions they generate bear higher complexity. This placement shows SNIP's strong Accuracy-Complexity balance for data without noise ($\gamma = 0$). To accentuate this advantage, Fig. 14(b) offers a deep dive into the Pareto Accuracy-Complexity performances of leading SR baselines on the *Feynman* dataset across a spectrum of target noise levels. Predictably, increased noise compromises the performance of all algorithms. Still, SNIP consistently holds a favorable position, indicating it generates more accurate and less complex expressions even with increased noise.

**Black-box:**    The study by SRBench (La Cava et al., 2021) delved into black-box problems, which were initially derived from OpenML[5] and later incorporated into the PMLB datasets (Olson et al., 2017). The intent behind assessing SR models on this dataset revolves around understanding how SR methods measure up against conventional well-known machine learning techniques, especially when faced with real-world, potentially noisy or sparse datasets. As delineated in Sec. E.3, the *Black-box* dataset used here comprises 57 regression datasets. In Fig. 13(**bottom**), we compare SNIP to the SRBench algorithms using this dataset. The figure shows how each algorithm performs in terms of accuracy, complexity, and inference time. The "*" before some names means they're SR methods; others are machine learning methods. From the figure, it's clear that SNIP does better than both the E2E transformer SR model and many other top SR methods. For accuracy on Black-box datasets, measured by the median $R^2$ score in SRBench, SNIP is ranked fourth among all methods. Its inference time on the Black-box dataset is similar to most other competing methods. Diving into complexity, SNIP outperforms many of its top-tier peers. Specifically, SNIP provides an average complexity score of $47.52$ on the Black-box datasets, which decisively outperforms its counterparts like Operon ($64.95$), SBP-GP ($639.19$), FEAT ($74.18$), EPLEX ($55.82$), and E2E ($82.78$). Further analysis of this balanced accuracy and complexity is also provided in Fig.6(b), which presents Pareto plots capturing both dimensions for the Black-box datasets. Consistent with earlier observations, SNIP holds its position on the first Pareto-front for this dataset. On the other hand, several competitors, including SBP-GP, FEAT, and EPLEX, are located on the secondary Pareto level, with E2E placing even further on the third. Such placements underscore their relative shortcomings in balancing accuracy with complexity, especially compared to SNIP. While GP-GOMEA shares the first Pareto-front with SNIP, its fitting accuracy is notably subpar, ranking 9th in Fig. 13–falling behind conventional ML methods like XGBoost and LGBM. So, out of all methods, Operon is the closest competitor to SNIP. Operon fits the data a bit better with a score of $0.933$ compared to SNIP 's $0.872$, but it offers more complex expressions with a score of $64.95$ against SNIP 's simpler $47.52$.

### E.5    Additional Results on the In-domain Synthetic Datasets.

**Detailed Results.**    We evaluate SNIP against the E2E transformer baseline on an in-domain synthetic validation set. This set consists of 400 equation examples following the data generation protocol from Sec. A. Functions uniformly vary across difficulty factors: input dimension $d \sim \mathcal{U}(1, d_{max})$, number of unary operators $u \sim \mathcal{U}(0, u_{max})$, binary operators $b \in \mathcal{U}(d-1, d+b_{max})$ where $d_{max} = 10$, $b_{max} = 4$, $u_{max} = 4$. We generated sequences of equation examples for each function by providing 200 input points $(x, y)$, and assessed prediction accuracy on another set of 200 test points. The fitting accuracy, denoted as $Acc(R^2 > 0.99)$ , is the proportion of solutions where the $R^2$ score exceeds 0.99. Additionally, the complexity is quantified as the number of nodes within the expression tree of the generated equations. Figure 15 presents a detailed comparison between SNIP and the E2E transformer baseline regarding fitting accuracy and the complexity of the derived equations on the in-domain dataset. The results demonstrate how the models' performance is affected by increasing formula complexity, as characterized by a higher number of operators and input dimensionality. Results show that as problem difficulty grows via more operators, SNIP maintains higher accuracy with a lower corresponding complexity increase compared to the E2E baseline.

**Ablation over Impact of Latent Space Optimization.**    To further demonstrate the impact of SNIP Latent Space Optimization (LSO), we conducted additional experiments using optimizers from the Nevergrad library (Rapin & Teytaud, 2018). The results on 400 held-out synthetic validation functions are shown in Table 3. We evaluate four configurations: (1) SNIP without LSO, (2) SNIP

---

[5]`https://www.openml.org/`

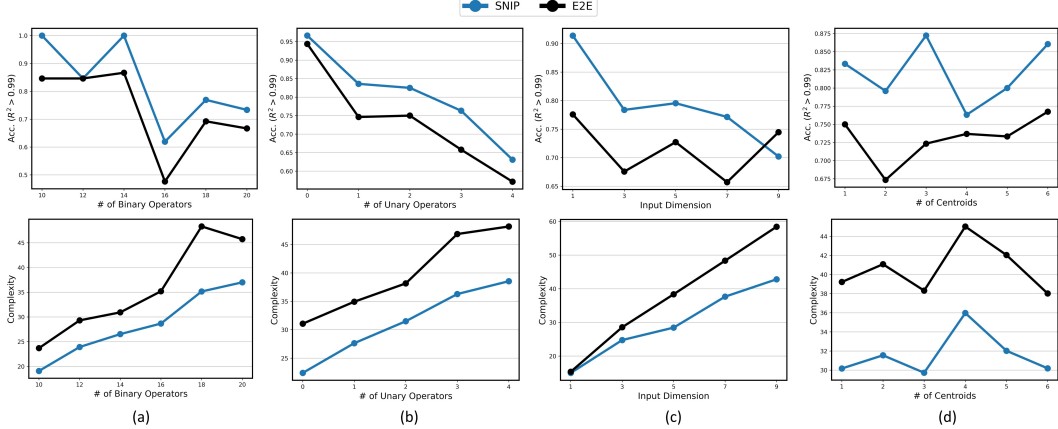

Figure 15: Detailed performance comparison of SNIP and the E2E Transformer baseline on 400 synthetic validation functions. Performance is measured by **Mean Accuracy (top)** and **Mean Expression Complexity (bottom)** across different levels of formula and input difficulties: **(a)** number of binary operators, **(b)** number of unary operators, **(c)** input dimension, and **(d)** number of input centroids. Mean accuracy reflects the percentage of solutions with $R^2 > 0.99$. Mean expression complexity quantifies the average prefix length of generated expressions.

with LSO using our employed Grey Wolf Optimizer (GWO), (3) SNIP with LSO using Nevergrad's NGOpt optimizer, and (4) SNIP with LSO using Nevergrad's TwoPointsDE optimizer. Table 3 shows the results of experiments, highlighting the mean $R^2$ score greater than $0.99$ and the mean complexity of the discovered equations. These metrics provide insight into the performance of the SNIP model with LSO using different optimization strategies. The results conclusively demonstrate the substantial gains provided by adding LSO to harness SNIP's latent space for symbolic regression. With LSO, the mean accuracy improves from $0.683$ to over $0.80$ regardless of the gradient-free optimization algorithm. The minor differences between optimizers can be attributed to variances in implementation and parameter tuning. Therefore, these additional experiments and analysis help demonstrate the significant benefits unlocked by performing LSO over SNIP's semantic and continuous latent representations. The substantial gains in accuracy underscore the importance of LSO as an integral component of our overall approach in using SNIP for SR.

Table 3: Performance comparison of SNIP with different LSO configurations.

| Model Configuration | $R^2 > 0.99$ | Complexity |
|---|---|---|
| SNIP w/o LSO | 0.683 | 28.43 |
| SNIP (LSO w/ GWO) | 0.820 | 29.95 |
| SNIP (LSO w/ NGOpt) | 0.805 | 30.21 |
| SNIP (LSO w/ TwoPointsDE) | 0.805 | 29.91 |