# OpenReview forum: "SNIP: Bridging Mathematical Symbolic and Numeric Realms with Unified Pre-training"
_ICLR.cc/2024/Conference — ICLR 2024 spotlight_

### Official Review · Reviewer_8R9V · 2023-10-18

**Soundness:** 4 excellent
**Presentation:** 4 excellent
**Contribution:** 3 good
**Rating:** 8
**Confidence:** 5

**Summary:**

This paper introduces SNIP, a multimodal pre-training method whose aim is to combine the symbolic and numeric properties of functions into a structured and meaningful latent space.
The authors show that the resulting embeddings can be used for a variety of downstream prediction tasks such as predicting the degree of convexity or upwardness of a function. They also combine their model with a decoder and a latent space optimiser to produce a powerful symbolic regression method.

**Strengths:**

Overall this a very enjoyable and valuable paper. I particularly appreciated:
- Novelty: although the paper builds on top of existing approaches, the CLIP-like training is a clever contribution which gives a lot of value to the paper.
- Impact: I believe this approach has a real potential to impact the way we perform symbolic tasks.
- Clarity: the paper is extremely pleasant to read. Although some parts lack details (see weaknesses), the paper is very well-written, easy to follow, and the figures are excellent.
- Content: I really enjoyed the experiments the authors present. The embeddings plot of Fig. 2 and interpolability plot of Fig. 5 give a lot of intuition, and the applications to cross-modal property prediction
- Experimental soundness: the authors put a lot of effort into evaluation. They rigorously benchmarked their model using SRBench (which many SR papers avoid still today), and have an extensive set of additional results in the appendix.
- Code availability: the authors provide an anonymous link towards their code.

**Weaknesses:**

In some parts, the authors are too hasty in their presentation of the technical details, missing out some important informations (see the questions below for details, and some other minor issues). Apart from this, I do not have many concerns about this paper.

**Questions:**

- There is a lack of details in some parts of the paper:
    - Finetuning (section 4): (i) It isn’t clear to me whether the properties (NCR and upwardness) are predicted based on the symbolic or numeric data. (ii) What is the dimension ? I don’t see how one can define upwardness for multi-dimensional functions so I assume D=1?
    - Interpolability plot (figure 5):  are these graphs qualitative plots, or are they actual functions in the latent space ? I would like the authors to comment on this.
- I’m not sure the fine-tuning task with 100 examples can be called few-shot. Few-shot usually refers to only a handful of examples. I would instead call this the “low data regime".
- In Fig 3, it looks like supervised model is going to catch up with SNIP… This is a bit problematic, ideally we would need to see results with more training examples to make sure SNIP remains better asymptotically. Authors the authors may argue that SNIP is much better in the low data regime, I don’t think this is the most important asset of SNIP, as we typically have infinite data in these kind of tasks anyway.
- I find it a bit weird to use a mapping network to convert the latent vector to a sequence. I understand that the authors want to mimic the cross-attention of the decode as in Kamienny et al, but intuitively, I would have instead tried using the latent vector as a “BOS” token, then use cross-attention with the output embeddings of the encoder (before they are pooled). This could be worth trying in the future.
- I don’t find the results section of sec 5.4.2 particularly well presented, especially in paragraphs “strogatz”, “black box” and “Feynman” where many numbers are presented which can directly be read from the graph — since the orderings of the models are similar on all three datasets, it would be better to have a more qualitative description extracting the key takeaways

---

> ### Author Response · Authors · 2023-11-18
> **Response to Reviewer 8R9V**
>
> **Details and Clarifications on section 4 and figure 5.**
> Thank you for the helpful questions - they will allow us to communicate these important details more clearly. Answer to your points:
> * In Sec 4, the input to the model is solely the symbolic equations, and the numeric properties (NCR, Upwardness) are predicted for these symbolic inputs. This aligns with the cross-modal nature of the tasks. As you noted, these properties are defined for 1D functions (D=1). We have updated the text to clearly state this detail.
> * The graphs in the interpolatability analysis (Figure 5) correspond to actual functions in the SNIP's latent space. For example, the blue point $Z^s_V$ corresponds to expression $f_s=x^2$, and the bottom-right orange point $Z^d_V$ corresponds to expression $f_d=\cos{(4 x)}+7$. When we interpolate between these points in the latent space (black point $Z_V^{int}$), the decoded expression $f_{int}=1.2\cos{(2.3x)}-0.9\sqrt{9-0.1x}+7$ exhibits a behavior between $f_s$ and $f_d$. This indicates the continuity of the latent space - small movements in the latent vectors result in semantically meaningful changes in the numerical behaviors of the decoded functions. This is a major advantage for symbolic regression, allowing us to transform the combinatorial search into a smoother semantic search by optimizing in the continuous latent space.
>
> **Mapping Network and the alternative suggestion.**
> Thanks for suggesting this insightful alternative approach. As noted correctly, one of the motivations for using the mapping network was to integrate with the pre-trained decoder weights from Kamienny et al. 2022, which have strong existing knowledge of expression generation. Cross-attending with encoder outputs (before pooling) could be an interesting approach to explore in future work. However, one potential challenge with this approach is that it would preclude the use of latent space optimization, which provides key benefits for symbolic regression: Performing optimization directly in the higher-dimensional encoder output space (e.g. N x demb = 200 x 512) is significantly more difficult than in SNIP's pooled 512-dim latent space. The continuity and semantics captured in SNIP's latent space allow an efficient yet meaningful search. Many existing neural symbolic regression methods do not incorporate optimization-based search. Those that do, tend to perform it after decoding, which limits the search capabilities. SNIP's main motivation for symbolic regression is to employ latent space optimization which integrates strong neural priors with the strengths of semantic search.
>
> ---
> > * In Fig 3, it looks like supervised model is going to catch up with SNIP… This is a bit problematic, ideally we would need to see results with more training examples to make sure SNIP remains better asymptotically. The authors may argue that SNIP is much better in the low data regime, I don’t think this is the most important asset of SNIP, as we typically have infinite data in these kind of tasks anyway.
> >
> Thanks for pointing this out. We agree that ideally, it would be preferable to show SNIP maintains its advantage asymptotically. However, it is important to consider that SNIP's pre-training phase uses only unlabeled data to learn general symbolic-numeric representations, unlike the supervised baseline model directly trained on labeled data for each specific task.  When evaluated on the downstream property prediction tasks in Sec 4, SNIP encounters the same limited labeled examples as the baseline. The fact that the specialized supervised model approaches SNIP’s performance with more data is expected, since it is tailored to the prediction task of this specific property. In fact, SNIP's value proposition is in its flexibility to efficiently adapt to new tasks thanks to its task-agnostic pre-trained representations. This capability makes it complementary to task-specific supervised models. For example, even for CLIP model, despite observing hundreds of millions of paired examples during pre-training, it achieves comparable performance to specialized models on image classification tasks, without necessarily outperforming them.
>
> ---
> > * I’m not sure the fine-tuning task with 100 examples can be called few-shot. Few-shot usually refers to only a handful of examples. I would instead call this the “low data regime".
> > * I don’t find the results section of sec 5.4.2 particularly well presented, especially in paragraphs “strogatz”, “black box” and “Feynman” where many numbers are presented which can directly be read from the graph — since the orderings of the models are similar on all three datasets, it would be better to have a more qualitative description extracting the key takeaways.
> >
> Thank you for your valuable suggestions. We have addressed each of the points you raised in the updated version.

---

> > ### Comment · Reviewer_8R9V · 2023-11-21
> > **Response**
> >
> > Thanks a lot for your detailed response, and congratulations on a really good submission!

---

> > > ### Author Response · Authors · 2023-11-22
> > > **Thank you!**
> > >
> > > Thank you very much for endorsing our work and for the helpful suggestions.

---

### Official Review · Reviewer_PqVT · 2023-10-27

**Soundness:** 3 good
**Presentation:** 3 good
**Contribution:** 4 excellent
**Rating:** 6
**Confidence:** 5

**Summary:**

The paper proposes SNIP, a joint encoder for numeric and symbolic mathematical data, -- i.e. functions and their values. SNIP consists of two transformer-based encoders, one for numeric sequences, and one for symbolic functions (including numeric pre-factors: e.g. 6.21cos(2.1x+3.25)). At the output of each encoder, an attention pooling mechanism converts the encoded sequence into a fixed-dimension embedding.

The numeric encoder architecture is very similar to Kamienny (2022). Floating point numbers are discretized into three tokens (following Charton 2022), D-dimensional vectors (represented as 3D tokens) are reduced to a single embedding using a feed-forward network (as in Kamienny), and sequences of N embeddings are fed, without positional encoding, into a 8 layer transformer with 16 heads and 512 dimensions.

The SNIP architecture is pre-trained on generated pairs of functions and their values, similar to the generated training sets used for end-to-end symbolic regression (Biggo, d'Ascoli, Kamienny). The authors use a symmetric contrastive loss, and show that the trained embeddings account for qualitative properties of the encoded functions, such as their growth and convexity (Figure 2, and appendix C ans D).

The model is then tested on two tasks:
* predicting qualitative properties of functions, by training a one layer classifier on top of a pre-trained model (either frozen or trainable)
* symbolic regression, by adding an autoregressive 16-layer transformer decoder at the output of the  pre-trained numeric encoder, and performing pre-factor fine-tuning of the predicted solution (i.e. tuning the constants in the functional expression) by minimizing mean square error over perturbed input data (noisy and/or subsampled).

For property prediction, the pre-trained model achieves high accuracy after fine-tuning on a small sample of functions (1000, to 10000).
For symbolic regression, the pre-trained model achieves state-of-the-art performance on the SRBench evaluation dataset. It outperforms better previous end-to-end approaches (Kamienny) and is comparable to the best evolutionary methods (Operon, SBP-GP).

**Strengths:**

Joint training on functions and their values is a promising idea, and the paper proposes a working architecture. The results on symbolic regression achieve a new state of the art for end-to-end transformer-based methods, on par with the best evolutionary tools.

The paper is clearly written, and the method is sufficiently described. Experiments are convincing.

**Weaknesses:**

* The architecture is complex and no ablation studies are provided. In particular, the model is larger than previous works on symbolic regression with transformers. SNIP has 8 and 16 layers in the encoder and decoder vs 6 and 6 in Biggio, and 4 and 8 in Kamienny. Without an ablation study, it is difficult to know if the improved performance is due to the new architecture, or is just an effect of scaling.

* Most evaluations focus on symbolic regression, which is also the pre-training task. Therefore, the claim that SNIP can be used on many mathematical tasks is not well-supported. To prove that the model can be applied to other tasks, evaluations on symbolic to numeric tasks, like function evaluation, or on pure symbolic or numerical computations (eg symbolic integration, linear algebra) would be needed.

* The test sets for symbolic regression are small, and somewhat limited. Feynman has 119 functions, many of them redundant, most of them very simple (first order approximations of physical laws). Strogatz has 14, all in low dimension. Evaluating from a larger test set of generated functions (held-out from the training set) would greatly improve the paper.

* There are no ablations on results. How does model performance scale with problem dimension, function complexity, the number of inputs, desired precision?

* Evaluation on function properties (section 4) is weak. The metrics used are redundant ($R^2=1-NMSE$), and not appropriate (what is the point of $R^2$ when only one number is predicted?). Additional metrics are needed, such as the number of correct predictions (e.g. model prediction within 1% of correct values).

* The comparison with unsupervised training (section 4) is inconclusive. The pre-trained models used 60 million examples, comparing them with an unsupervised model trained on 10,000 is unfair. In fact, the 87% accuracy achieved by the unsupervised model after only one million examples is quite surprising.

**Questions:**

**Section 3 - encodings**
* Alternative encodings for real numbers been proposed in previous works. Biggio feeds the floating point representations into the embedding, Becker et al. propose the two-bit encoding (https://arxiv.org/abs/2307.12617). Recently (after the submission deadline), Golkar et al. introduced XVAL (https://arxiv.org/abs/2310.02989), an improvement over the three token embedding. These alternatives should be referenced in the paper (either in section 3 or in the related works). A discussion of their relative merits would be useful.
* In several previous works,, symbolic functions are encoded as "skeletons", where pre-factors are replaced by constant tokens, e.g. CST cos(CST x + CST) instead of  6.21 cos (2.01 x + 3.25). Could they be used in SNIP, instead of encoding prefactors (as 3 token sequences)?
* Can you provide dataset sizes, notably the number of pre-training examples, in the main?
* Section 3.1 p.3: the base 10 floating point encoding, using three tokens per numbers, was introduced in Charton (2022)
* Section 3.2 p.4: the prefix enumeration of trees was introduced in Lample (2020)

**Section 4 - predicted properties**
* $R^2$ makes little sense when measuring scalar quantities. Also, $R^2=1-MSE$, so the columns in table 1 are redundant. Could you add scalar accuracy metrics, such as the percentage of test examples predicted within a certain tolerance of their actual values?
* Add the chance level for the indicators in table 1.
* The comparison between pre-trained and unsupervised model (figure 3) is unsatisfactory: it overlooks the fact that the pre-trained model already saw 60 million examples. The analysis is interesting but needs to be improved.
* Several properties were measured in this evaluation, but only average results are presented. Can detailed results for each property be provided? (maybe in the appendix)

**Section 5 - Symbolic regression**
* The decoder is significantly larger that what was used in previous works (16 layers, vs 8 in Kamienny and 6 in Biggio), why is this so? Can you provide ablation results for different architecture choices (layers, but also dimensions and heads)?
* Figure 11.a, in Appendix E, strongly suggests that SNIP is learning different (independent?) models for different dimensions D. Is there a benefit in training SNIP on many dimensions at the same time? Or should different models be trained on different dimensions?
* Please provide results on a held-out dataset of generated functions, and scaling results for different problem dimensions (D), function complexity (e.g. family of operators used), number of input (N), precision achieved (in the MSE sense).
* The late space optimisation experiments would benefit from a more detailed description in the main (perhaps at the expense of the introduction, which is a bit repetitive). In particular, the impact of LSO on model performance should be presented and discussed.
* A gradient-free library like Nevergrad (https://facebookresearch.github.io/nevergrad/) could be used to provide better perpective on the potential impact of LSO. At present, only one algorithm is used, with little justification.

---

> ### Author Response · Authors · 2023-11-18
> **Response to Reviewer PqVT (Part1)**
>
> ### **Encodings**:
> **Alternative numeric encodings.**
> We appreciate the reviewer highlighting recent alternate number encodings like Golkar et al. 2023, and Becker et al. 2023. We agree that discussing the merits of emerging schemes would enrich the background and have acknowledged them in the updated related work. However, analyzing tradeoffs between encoding schemes is not the core focus of our work, and needs further exploration which can be a potential future work. These potential directions will certainly involve challenges and may demand rigorous experimentation to ascertain their efficacy. SNIP's primary contribution lies in pioneering cross-modal pre-training between mathematical symbols and numerics. This unified learning of symbolic and numeric representations is orthogonal and complementary to the choice of encoding scheme.
>
> **Skeleton vs. Prefactors**
> We appreciate the reviewer's insightful question. We actually explored this direction initially. However, our preliminary experiments showed limitations of solely using equation skeletons without numeric constants. Specifically, pre-training with just skeletons cannot effectively imprint insights about a function's unique numeric behavior, as each skeleton can manifest vastly different numeric behaviors based on different constants. This hindered capturing cross-modal similarities in the learned representations and led to subpar performance in tasks where this numeric pattern is critical such as prediction of numeric properties or symbolic regression. For example, the convexity of a simple function like $f(x) = C x^2$ depends heavily on the choice of constant $C$ as it is fully convex for $C>0$, and non-convex for $C<0$.
>
> **Suggested modifications on dataset size and references.**
> Thanks for reviewing these details. The total number of pre-training examples used for SNIP is around $60M$. This information is now added to the main manuscript for clarity. We have also updated the references for accuracy in the new version.
>
> ### **Property Prediction**:
> **Evaluation metrics, chance level, and results for other properties.**
> We thank the reviewer for these valuable suggestions. To improve the evaluation on function properties we have done the following steps as suggested:
> 1) To avoid redundancy, we have removed the column for $R^2$, and only provide NMSE.
> 2) As suggested, we define an accuracy metric, based on the closeness of the predicted values to their corresponding true values. To this end, we first normalize the true values for all of the tasks to the range of $(0, 1)$. Subsequently, we define the accuracy within absolute tolerance of $\tau$ as $Acc_{\tau} = \frac{1}{N_{t}} \Sigma^{N_{t}}_{i=1} \mathbb{1} \[ |\hat{p}_i - p_i| \leq \tau \] $, where $p_i$ and $\hat{p}_i$ are the true and predicted values for $i$-th example. This metric perceives a prediction as accurate if the predicted value is within tolerance (here, $\tau =0.1$) of the normalized values.
>
> 3) For NMSE, the chance level can be considered to be $NMSE=1$ as it corresponds to the predicting average of property without considering input values. For the newly defined accuracy metric, we also consider the base (chance level) as the accuracy achieved when assuming $\hat{p}_i = \bar{p}$ for all examples. We have reported the chance level for each property in the table below.
>
> 4) We have updated the results for the NCR and Upwardness metric in Sec 4.2 and Tab 1, and to be comprehensive, we have also added the results for all properties in App C.2 as suggested. For your reference, here are the updated results:
>
> | Model             | NCR |                | UP |                | Avg. y  |                | Oscillation  |                |
> |-------------------|---------------------|----------------|------------|----------------|-------------|----------------|------------------|----------------|
> |                   | ↓ NMSE              | ↑ $Acc_{0.1}$  | ↓ NMSE     | ↑ $Acc_{0.1}$  | ↓ NMSE      | ↑ $Acc_{0.1}$  | ↓ NMSE           | ↑ $Acc_{0.1}$  |
> | base        |                     | $2.4$%  |            | $22.4$% |             | $52.4$% |                  | $23.1$% |
> | Supervised        | 0.5299              | 56.5  %        | 0.5356     | 56.3  %        | 1.0406      | 49.3  %        | 0.3079           | 75.2  %        |
> | SNIP (frozen)     | 0.0731              | 86.1  %        | 0.0540     | 84.7  %        | 0.4532      | 64.5  %        | 0.0683           | 92.6  %        |
> | SNIP (finetuned)  | **0.0683**          | **92.1**  %        | **0.0400** | **90.1**  %        | **0.4074**  | **67.7**  %        | **0.0581**       | **92.6**  %        |
>
> We observe that SNIP predictions have better performance compared with the supervised model in both metrics. The chance level depends on the distribution of values for each property. Also, different tasks have various difficulties, and SNIP's performance can be better in some cases (e.g. NCR) compared to more challenging properties (e.g. Avg y).

---

> > ### Author Response · Authors · 2023-11-18
> > **Response to Reviewer PqVT (Part2)**
> >
> > ### **Property Prediction (Continue)**
> > ---
> > > * The comparison with unsupervised training (section 4) is inconclusive. The pre-trained models used 60 million examples, comparing them with an unsupervised model trained on 10,000 is unfair. In fact, the 87% accuracy achieved by the unsupervised model after only one million examples is quite surprising.
> > > * The comparison between pre-trained and unsupervised model (figure 3) is unsatisfactory: it overlooks the fact that the pre-trained model already saw 60 million examples. The analysis is interesting but needs to be improved.
> > >
> > We appreciate the reviewer's comment. Regarding the fairness of the comparison, it's crucial to note that during its pre-training phase, SNIP was not exposed to any labeled data. Instead, it was trained to learn mutual symbolic-numeric similarities in representations, akin to the approach used by CLIP in image-language pre-training. This process was inherently unsupervised.
> > In Sec 4, when SNIP is employed to cross-modal property prediction tasks, it encounters the same number of labeled examples as the baseline model. The fact that the supervised baseline model approaches SNIP's performance with more training data (1M examples) is reasonable and can be expected since this model is specialized only in the prediction task for each property. However, SNIP's value lies in its flexibility - the pre-trained representations can be efficiently adapted to new tasks, unlike supervised baseline models trained from scratch for each task. SNIP demonstrates this advantage, similar to models like CLIP, which, despite being exposed to hundreds of millions of paired samples during pre-training, achieves comparable performance to task-specific models in image classification without necessarily outperforming them.
> > To further clarify this for the readers, we have included an additional analysis on this comparison in Sec 4.
> >
> > ### **Symbolic Regression**:
> > ---
> > > * The architecture is complex and no ablation studies are provided. In particular, the model is larger than previous works on symbolic regression with transformers. SNIP has 8 and 16 layers in the encoder and decoder vs 6 and 6 in Biggio, and 4 and 8 in Kamienny. Without an ablation study, it is difficult to know if the improved performance is due to the new architecture, or is just an effect of scaling.
> > > * The decoder is significantly larger that what was used in previous works (16 layers, vs 8 in Kamienny and 6 in Biggio), why is this so? Can you provide ablation results for different architecture choices (layers, but also dimensions and heads)?
> > >
> > We appreciate the reviewer's comments about analyzing SNIP's architecture choices. The reviewer is correct that the SNIP encoder uses more layers (8) compared to previous works like Kamienny (4 layers) and Biggio (6 layers). This design choice was motivated by SNIP's focus on learning rich joint representations, for which added encoder capacity could be beneficial. However, the reviewer's statement about SNIP having a larger 16 layer decoder is not correct. As noted in the manuscript (pg 7, Sec 5.1, p.3), the 16 layer decoder is directly initialized from the Kamienny et al. (2022) model. We use a mapping function to leverage their pre-trained weights. So the decoder architecture to use SNIP for symbolic regression matches previous work.
> >
> > In response to the reviewer's comment, we have begun ablation studies on SNIP encoder variants with 4 layer encoders, and 8 attention heads. However, this is a very time-consuming experiment and full comparative results on symbolic regression tasks would require not only pre-training the new SNIP variant from scratch but also re-training a decoder on top of the new SNIP variant for symbolic regression. We will include the results in the camera-ready version as the corresponding experiments are too long to be run during the rebuttal period. Additionally, we would like to note that the key value of SNIP lies in its novel task-agnostic pre-training approach rather than architectural innovations. So determining the exact sources of performance gains is secondary to demonstrating SNIP's capabilities, which we've aimed to do through strong comparative benchmarking.

---

> ### Author Response · Authors · 2023-11-18
> **Response to Reviewer PqVT (Part3)**
>
> ### **Symbolic Regression (Continue)**
> ---
> > * Figure 11.a, in Appendix E, strongly suggests that SNIP is learning different (independent?) models for different dimensions D. Is there a benefit in training SNIP on many dimensions at the same time? Or should different models be trained on different dimensions?
> >
> We appreciate the reviewer's insightful question. Fig 11 does illustrate that SNIP learns clustered representations based on the input dimension. This indicates that the model successfully captures similarities between functions of the same dimensionality during pre-training. The benefit of training a single SNIP model on a range of input dimensions (up to 10D in our case) is that it allows flexible application on datasets with varying dimensionality without needing to train specialized models per dimension. This is a more efficient approach than training separate SNIP variants for each input dimension. Importantly, Fig 11.b and 11.c also show meaningful structure within each dimension's cluster based on the symbolic-numeric essence of the functions. So while distinct dimension representations are learned, the pre-trained encodings still embed cross-modal knowledge.
>
> ---
> > * The test sets for symbolic regression are small, and somewhat limited. Feynman has 119 functions, many of them redundant, most of them very simple (first order approximations of physical laws). Strogatz has 14, all in low dimension. Evaluating from a larger test set of generated functions (held-out from the training set) would greatly improve the paper.
> > * Please provide results on a held-out dataset of generated functions, and scaling results for different problem dimensions (D), function complexity (e.g. family of operators used), number of input (N), precision achieved (in the MSE sense).
> > * How does model performance scale with problem dimension, function complexity, the number of inputs, desired precision?
> >
> Thank you for the suggestion. In the original manuscript, we focused evaluations on SRBench given its recognition as the gold standard benchmark and the inclusion of leading GP baselines. However, in response to the reviewer's comment, we have additionally tested SNIP on the held-out in-domain datasets, generated following the data generation protocol in Sec 3.4, consisting of 400 validation functions with different levels of difficulty: number of binary operators ($b$), number of unary operators ($u$), input dimension ($d$), and number of input data centroids ($c$). The detailed results in Fig 15 of App E.5 demonstrate how the models' performance is affected by increasing formula complexity, as characterized by a higher number of operators and input dimensionality. We have provided these additional results in the appendix due to the page limitation, and that SRBench datasets, including Feynman, Strogatz, and Blackbox, constitute the most widely used and challenging open-source benchmark for symbolic regression. In fact, many recent pre-trained models (Biggio et al. 2021, Kamienny et al. 2022) have shown even weaker performance on SRBench compared to held-out synthetic datasets from their pre-training distribution. This highlights the greater difficulty of generalizing to out-of-domain datasets like SRBench.

---

> > ### Author Response · Authors · 2023-11-18
> > **Response to Reviewer PqVT (Part4)**
> >
> > ### **Symbolic Regression (Continue)**
> > ---
> > > * The late space optimisation experiments would benefit from a more detailed description in the main (perhaps at the expense of the introduction, which is a bit repetitive). In particular, the impact of LSO on model performance should be presented and discussed.
> > > * A gradient-free library like Nevergrad (https://facebookresearch.github.io/nevergrad/) could be used to provide better perpective on the potential impact of LSO. At present, only one algorithm is used, with little justification.
> > >
> > We appreciate the reviewer's recommendations to strengthen the description and analysis of Latent Space Optimization (LSO) in our work. Based on your feedback:
> > 1. We have expanded the details on LSO in the main body of the paper to better convey its integral role in harnessing SNIP's capabilities for symbolic regression. We clarify how LSO transforms the combinatorial search problem into an efficient optimization within SNIP's continuous, semantically meaningful latent space.
> > 2. To further demonstrate the impact of LSO and the choice of gradient-free algorithm, we conducted additional experiments using the `Nevergrad` library's gradient-free optimizers as suggested. Results on 400 held-out synthetic validation functions confirm that adding LSO on top of SNIP's representations provides significant gains in the fitting accuracy of generated expressions, with minor differences between various optimizers. We can see that the currently employed GWO algorithm (with search strategies close to those of PSO) results in a slightly better performance compared to the `NGOpt` and `TwoPointsDE` algorithms from the Nevergard library. Please note that there are minor differences in the implementation of our algorithm compared to using Nevergrad ready-to-use optimizers, but we have tried our best to perform a fair comparison. We have included these additional experiments in App E.5.
> >
> >
> > | Model Configuration        | $R^2>0.99$ | Complexity |
> > |----------------------------|-----------------|-----------------|
> > | SINP w/o LSO               | 0.683           | 28.43           |
> > | SNIP (LSO w/ GWO)          | 0.820           | 29.95           |
> > | SNIP (LSO w/ NGOpt)        | 0.805           | 30.21           |
> > | SNIP (LSO w/ TwoPointsDE)  | 0.805           | 29.91           |
> >
> > ### **Others**:
> > ---
> > > * Most evaluations focus on symbolic regression, which is also the pre-training task. Therefore, the claim that SNIP can be used on many mathematical tasks is not well-supported. To prove that the model can be applied to other tasks, evaluations on symbolic to numeric tasks, like function evaluation, or on pure symbolic or numerical computations (eg symbolic integration, linear algebra) would be needed.
> > >
> > Thank you for the feedback. We would like to clarify that SNIP is not inherently tailored for symbolic regression during its pre-training phase. Instead, SNIP represents a task-agnostic pre-training approach that is designed to learn cross-modal similarities in representations. This feature is crucial in understanding SNIP’s broader applicability beyond the scope of symbolic regression. In the manuscript, we have shown the application of SNIP in the context of numeric-to-symbolic generation task, known as symbolic regression. Furthermore, in Sec 4, we have showcased SNIP’s utility in symbolic-to-numeric property prediction tasks.  Here, SNIP's ability to draw upon numeric insights from symbolic encodings is highlighted, reinforcing the model's versatility. However, it’s important to stress that these two downstream tasks – symbolic regression and property prediction – were selected as initial proof-of-concept applications to illustrate the effectiveness and potential of SNIP's pre-training strategy in handling cross-modal (numeric-to-symbolic and symbolic-to-numeric) reasoning. We acknowledge and agree with the reviewer’s point that extending evaluations to a wider range of tasks, such as function evaluation, symbolic integration, etc., would provide a more comprehensive demonstration of SNIP’s capabilities. This is indeed an exciting and promising direction that we are keen to pursue in future research endeavors, as mentioned in Sec 6 on future work. We believe that SNIP’s innovative approach holds significant potential for a variety of mathematical applications.

---

> > > ### Comment · Reviewer_PqVT · 2023-11-22
> > >
> > > Thank you very much for the detailed answers. I will take them into account during the discussion with the area chair and other reviewers.

---

> > > > ### Author Response · Authors · 2023-11-22
> > > > **Thank you!**
> > > >
> > > > Thanks a lot for your detailed and constructive questions and suggestions. Please feel free to let us know if you have more questions or concerns.

---

### Official Review · Reviewer_RHHe · 2023-10-29

**Soundness:** 3 good
**Presentation:** 3 good
**Contribution:** 3 good
**Rating:** 8
**Confidence:** 4

**Summary:**

The paper introduces SNIP, a Symbolic Numeric Integrated Pre-training approach, meant to unify numeric and symbolic domains via contrastive learning. The proposed method follows the rich literature on multi-modal pre-trained models whose main goal is to combine different modalities such that their representations are appropriately linked and correlated in the latent space.
SNIP is based on two encoders, one for numerical data and one for symbolic expressions. Contrastive learning is used to unify the resulting representations. Once this pre-training phase is completed, the numerical encoder can be extracted and combined with a symbolic decoder responsible for predicting the symbolic expression associated with the input numerical values, a.k.a. performing end-to-end symbolic regression. Thanks to the rich structure acquired by the numerical encoder during pre-training, latent space optimisation can be performed to enhance the results of symbolic regression. The results show that SNIP's encoders contain cross-domain information about symbolic and numerical inputs and that symbolic regression benefits from the pre-training phase.

**Strengths:**

In terms of clarity the paper is generally well written and easy to follow.

In the context of end-to-end symbolic regression performed by Transformer models, the problem of unifying numerical and symbolic representations is very relevant. As such, I believe SNIP represents an interesting step in this direction.

The experiments in Section 4 and Appendix C effectively demonstrate that the single encoders' representations contain cross-domain information, e.g. the symbolic encoder's latent space is informative about numerical properties associated with the evaluations of the input expression. In addition, the example in Fig. 5 seems to suggest that the latent space possesses a continuous structure, in the sense that nearby points can be decoded into expressions that have a similar numerical evaluations.

The results in Section 5 show that the proposed approached outperforms E2E, which is the closest baseline deep learning model for symbolic regression. The model performs comparably with state-of-the art genetic-programming-based symbolic regression methods.

**Weaknesses:**

- Some parts could be better explained and more details should be provided. For example, in Section 4 it is not clear how the task is performed. Looking at the Appendix, it looks like the prediction of the NCR and Function Upwardness is made by extracting the embedding from the symbolic encoder only as the goal would be to assess whether the symbolic representations contain numerical information as well. If my interpretation is correct, I believe this should be explained more clearly in the main body.
Another source of confusion is the way symbolic regression is performed. In particular, it is not clear to me if the decoder is trained or not. Are the authors keeping it frozen and training only the mapping network?

- The experiments performed in Section 4 are conducted on 1-d symbolic expressions. Although this setting is already insightful, I would be curious to see whether similar tests can be conducted on higher-dimensional expressions.

- While Transformer-based approaches to symbolic regression provide advantages in terms of inference time compared to genetic programming approaches, they may suffer from some of the limitations characterizing standard language models. One such limitation could be that they may hallucinate and output a symbolic expression that does not fit the numerical data at all. I would have liked to see some investigations on this aspect, for example an analysis on the extent to which output symbolic expressions are numerically aligned with the input dataset.

**Questions:**

See weaknesses part.

---

> ### Author Response · Authors · 2023-11-18
> **Response to Reviewer RHHe**
>
> **Details and Clarifications on the requested sections.**
> Thank you for the helpful questions - they will allow us to communicate these important details more clearly. Answer to your points:
> * **Section 4:** Yes, you're correct. the input to the model is solely the symbolic equations, and the numeric properties (NCR, Upwardness) are predicted for these symbolic inputs. This aligns with the cross-modal nature of the tasks. We have updated the text to clearly state this detail.
> * **Section 5:** For the symbolic regression, the decoder is initialized from the pre-trained weights of Kamienny et al. (2022), and then trained with the mapping network for the expression generation task (with cross-entropy loss). Details of symbolic regression training are provided in pg 20, App E.1, p.4. We have also included a summary in the main text for clarity.
>
> ---
> > * The experiments performed in Section 4 are conducted on 1-d symbolic expressions. Although this setting is already insightful, I would be curious to see whether similar tests can be conducted on higher-dimensional expressions.
> >
> We appreciate the reviewer's insightful question. The current properties we defined and analyzed are primarily intended for 1D examples, as a proof of concept to showcase SNIP's capabilities for cross-modal reasoning between symbolic equations and numeric data patterns. Expanding this analysis to higher-dimensional problems, however, would require defining new properties relevant to such contexts, as most current properties are focused on 1D scenarios. If the reviewer has any further thoughts or suggestions regarding insightful multi-dimensional properties worth exploring, we would highly appreciate the input. Higher-dimensional analysis on new properties would also be time-consuming as we'd need to train the three model variants to predict these new properties in higher-dimensional contexts.
>
> ---
> > * While Transformer-based approaches to symbolic regression provide advantages in terms of inference time compared to genetic programming approaches, they may suffer from some of the limitations characterizing standard language models. One such limitation could be that they may hallucinate and output a symbolic expression that does not fit the numerical data at all. I would have liked to see some investigations on this aspect, for example an analysis on the extent to which output symbolic expressions are numerically aligned with the input dataset.
> >
> Thanks for raising this very important point. As the reviewer noted correctly,  most of the Transformers neural symbolic regression models that are trained on large-scale synthetic datasets (e.g., Kamienny et al. 2022, and Biggio et al. 2021) struggle to generate accurate symbolic expressions due to their training objective (cross-entropy loss) being misaligned with equation discovery goals ( fitting data well). They rely on techniques like beam search and constant refinement to mitigate this, but these are not always sufficient. SNIP, however, targets this issue innovatively. Unlike traditional models that depend solely on Transformer logits and post-decoding search strategies, SNIP leverages latent space optimization (LSO) to integrate neural priors with semantic search with the objective of achieving higher fitting accuracy (e.g., $R^2$ score). As shown in Fig 5, SNIP's latent space provides both interpolatability and continuity, where minor adjustments in semantic latent vectors lead to meaningful changes in the numerical behaviors of decoded functions. This is a strong foundation for symbolic regression, allowing us to transform the combinatorial search into a smoother semantic search by optimizing in the continuous latent space.
>
> Based on the reported results in Sec 5.4 and Fig 6, as well as App E.4, we observe that SNIP achieves higher average $R^2$ scores compared to prior neural symbolic regression models. This indicates SNIP's output equations exhibit stronger numerical alignment with the input data. We would also like to note that to improve the numerical alignment of the generated equations, the candidate equations (generated from the decoder) also undergo a constant refinement stage (similar to Kamienny et al. 2022) which optimizes their coefficients based on the given dataset using BFGS optimization. By leveraging both LSO with $R^2$ objective and this refinement strategy, SNIP is able to closely fit the training data.

---

> > ### Author Response · Authors · 2023-11-22
> > **Thank you and follow up**
> >
> > Dear reviewer,
> > Thank you again for your valuable review. As the discussion period deadline is approaching, please feel free to let us know if our responses have addressed your concerns or if you have any additional questions or concerns.

---

> > ### Comment · Reviewer_RHHe · 2023-11-23
> >
> > Thank you for your answers. My concerns have been addressed and I will keep my score.

---

### Official Review · Reviewer_7mJN · 2023-10-30

**Soundness:** 3 good
**Presentation:** 3 good
**Contribution:** 3 good
**Rating:** 8
**Confidence:** 4

**Summary:**

The paper proposes a new model for pre-training an embedding space for mathematical expressions. The obtained embeddings aim at encoding both symbolic and numerical properties of the corresponding expressions.  The paper introduces a dual encoder-scheme to learn such joint embeddings, inspired by similar schemes in multi-modal representation learning.
The paper focuses on two tasks: property prediction and symbolic regression.

For property prediction, a simple extension of the embedding architecture is proposed, where an additional head is exploited to predict numeric and symbolic properties of the input equations. The results show that pretrained embeddings (possibly, fintuned) provide a much better prediction in terms or R^2 and NMSE.

For symbolic regression, a much more elaborate scheme is proposed. The existing encoding architecture is mapped with a pretrained-decoder using a technique similar to ClipClap. Second, an inference latent space optimization process is exploited to exploit the interpolability of the pertained latent space. The results show that SNIP is in the Pareto front for all the three investigated datasets.

**Strengths:**

The paper is well written and easy to follow.

The method is simple, sound and elegant.

Generating a large number of expressions automatically is easy: these settings are the ones that may benefit more from a pretraining strategy.

**Weaknesses:**

From the current shape of the paper, it is a little bit hard to investigate the novelty. The majority of techniques are inspired by similar techniques in multi-modal representation learning, and it is not clear whether there has been similar endeavors in the context of mathematical expressions representation learning or closer domains.

**Questions:**

1) It is unclear whether there are other methods that exploit pretraining for mathematical expressions, or other semi-supervised approaches. In both the comparison, it is clear that some of the methods have been exposed to a very different amount of (unsupervised) data. So one may wonder whether there are other methods of exploiting such data than the one proposed by the authors.

2) In order to apply the proposed method to symbolic regression, a LSO process is added on top of the standard training. It is unclear what is the impact of such step on the quality of the generation and whether competitors employ similar strategies and to which extent. Do the authors have performed an ablation study or have intuitions about the impact of the LSO?

---

> ### Author Response · Authors · 2023-11-18
> **Response to Reviewer 7mJN**
>
> **Novelty of SNIP and discussion on comparisons.**
> Thank you for the comments and thoughtful questions. Please find our answers below.
> * Pre-training in Mathematical Expressions: While our techniques draw inspiration from multi-modal representation learning, prevalent in domains like vision-language, their employment to mathematical expressions is unique. Existing Transformer-based models in this domain have been successful in tasks like differential equations and function integration but fall short in integrating symbolic and numeric data insights. Recent Transformer models in symbolic regression, like Kamienny et al. (2022) and Biggio et al. (2021), also only focus on mapping numeric data to mathematical expressions in a supervised, task-specific manner. To the best of our knowledge, **our work is the first approach that integrates these domains through a unified, task-agnostic pre-training**. This approach is distinct from existing methods and provides a novel perspective in mathematical representation learning.
> * Cross-modal Property Prediction Comparisons: During pre-training, SNIP only sees unlabeled data, learning mutual symbolic-numeric similarities in an unsupervised way. When later evaluated on downstream tasks, SNIP encounters the same limited labeled data as baseline models. The comparisons demonstrate SNIP's ability to efficiently adapt to new cross-modal prediction tasks thanks to pre-training - a valuable complement to specialized and task-specific supervised models.
> * Symbolic Regression Comparisons: In the SR comparisons, most of the baselines are based on Genetic Programming which does not use any prior knowledge, as they only see one dataset. Transformer-based baselines like Kamienny et al. (2022) employ supervised training on large synthetic datasets for the numeric-to-symbolic generation task of SR. As noted, SNIP's pre-training is unsupervised, focused on learning general symbolic-numeric representations like CLIP learns visual-textual relationships. For the SR task, we follow a similar strategy as Kamienny et al. (2022) to generate data and fine-tune the symbolic decoder in a supervised manner. Subsequently, we exploit SNIP's prior knowledge combined with LSO for enhanced equation generation.
>
> We hope our response addresses your concern. Please do let us know if we have correctly interpreted your concern or if further clarification is needed.
>
> **Impact of LSO in the use of SNIP for symbolic regression.**
> Thank you for pointing this out. To clarify, existing Transformer neural symbolic regression models that are trained on large-scale synthetic datasets struggle to generate accurate symbolic expressions due to their training objective (cross-entropy loss) being misaligned with equation discovery goals ( fitting data well). They rely on techniques like beam search to mitigate this, but these are not always sufficient. SNIP, however, targets this issue innovatively. Unlike traditional models that depend solely on Transformer logits and post-decoding search strategies, SNIP leverages latent space optimization (LSO) to integrate neural priors with semantic search. As shown in Fig 5, SNIP's latent space provides both interpolatability and continuity, where minor adjustments in latent vectors lead to meaningful changes in the numerical behaviors of decoded functions. This is a strong foundation for symbolic regression, allowing us to transform the combinatorial search of mathematical equations into a smoother semantic search by optimizing in the continuous latent space. To the best of our knowledge, none of the prior works in this domain have employed such a latent space search strategy. In fact, LSO is an integral component of our method when used in SR. It enables generating equations that leverage the prior knowledge from SNIP pre-training, while also fitting the dataset well.
>
> In response to the reviewer's comment, we conducted additional ablation experiments to further illustrate the impact of LSO in our method (on 400 held-out synthetic validation functions). The results (please see the table) show that performing LSO (with the objective of fitting accuracy) on the SNIP's rich representation space significantly improves the fitting performance of the generated equations compared to the variant without LSO, while maintaining the same level of complexity. We have included this study and additional discussion on the impact of LSO in App E.5.
>
> | Model Configuration        | $R^2 > 0.99$ | Complexity |
> |----------------------------|----------------------|----------------------|
> | SNIP w/o LSO          | 0.683       | 28.43       |
> | SNIP (w/ LSO)   | 0.820       | 29.95       |

---

> > ### Author Response · Authors · 2023-11-22
> > **Thank you and follow up**
> >
> > Dear reviewer,
> > Thanks you again for your valuable review. As the discussion period deadline is approaching, please let us know if we have been able to address your concerns, or if you have any additional questions or concerns.  Thank you!

---

### Author Response · Authors · 2023-11-18
**General Response to reviewers**

We sincerely thank all the reviewers for dedicating their time and expertise to review our manuscript. Your constructive feedback has been instrumental in enhancing our paper. We have provided a detailed response to each comment/question and have uploaded the revised manuscript and appendix with the changes highlighted in blue. Please do feel free to let us know if you have any further questions.

---

### Meta-Review · Area_Chair_erdt · 2023-12-09

**Metareview:**

The authors propose pre-training for mathematical expressions via multi-modal contrastive learning that jointly encodes the symbolic and numerical parts of the expressions via a dual encoder-scheme. The tasks considered are property prediction and symbolic regression.    The proposed method is then used for a qualitative assessment of the learned embeddings and for downstream symbolic regression tasks (adding a decoder). It results competitive w.r.t. recent baselines on these benchmarks.

In their reviews, reviewers appreciated the contribution of the paper, the joint pre-training for its simplicity, the good results on SRBench and the clarity of exposition. However, they also highlighted some missing details from the presentation. Authors provided most of them in the revised version of the paper. Additional discussion included: one reviewer also highlighting the limitation of being focused to unidimensional symbolic regression cases, and another reviewer questioning the lack of ablations for such a large pipeline. Authors partially answered in the rebuttal (and increased one reviewer's score).

The paper is accepted and authors are greatly encouraged to address the feedback of the reviewers (especially PqVT) in the camera-ready.

**Justification For Why Not Higher Score:**

The paper's contribution is essentially engineering a learning pipeline via contrastive learning for mathematical expressions and piggybacks on several recent advancements in multi-modal pre-training.

**Justification For Why Not Lower Score:**

The paper could be accepted as poster. One reason to have it as a spotlight, however, is to expose it beyond the symbolic regression community, but I am not sure that these embeddings can be truly useful elsewhere.

---

### Decision · Program_Chairs · 2024-01-16

Accept (spotlight)